



# Vegetation, ground cover, soil, rainfall simulation, and overland flow experiments before and after tree removal in woodland-encroached sagebrush steppe: the hydrology component of the Sagebrush Steppe Treatment Evaluation Project (SageSTEP)

**C. Jason Williams[1], Frederick B. Pierson[2], Patrick R. Kormos[3], Osama Z. Al-Hamdan[4], and Justin C. Johnson[1,5]**

[1] Southwest Watershed Research Center, USDA Agricultural Research Service, Tucson, AZ, USA

[2] Northwest Watershed Research Center, USDA Agricultural Research Service, Boise, ID, USA

[3] Colorado Basin River Forecast Center, USDC National Oceanic and Atmospheric Administration – National Weather Service, Salt Lake City, UT, USA

[4] Department of Civil and Architectural Engineering, Texas A&M University-Kingsville, Kingsville, TX, USA

[5] School of Natural Resources and the Environment, University of Arizona, Tucson, AZ, USA

*Correspondence to:* C. Jason Williams (jason.williams@usda.gov)

**Abstract.** Rainfall simulation and overland-flow experiments enhance understanding of surface
hydrology and erosion processes, quantify runoff and erosion rates, and provide valuable data for developing and testing predictive models. We present a unique dataset (1021 experimental plots) of rainfall simulation (1300 plot runs) and overland flow (838 plot runs) experimental plot data paired with measures of vegetation, ground cover, and surface soil physical properties spanning point to hillslope scales. The experimental data were collected at three sloping sagebrush
(*Artemisia* spp.) sites in the Great Basin, USA, each subjected to woodland-encroachment and with conditions representative of intact wooded-shrublands and 1-9 yr following wildfire, prescribed fire, and/or tree cutting and shredding tree-removal treatments. The methodologies applied in data collection and the cross-scale experimental design uniquely provide scale-dependent, separate measures of interrill (rainsplash and sheetflow processes) and concentrated
overland-flow runoff and erosion rates along with collective rates for these same processes combined over the patch scale (tens of meters). The dataset provides a valuable source for developing, assessing, and calibrating/validating runoff and erosion models applicable to diverse plant community dynamics with varying vegetation, ground cover, and surface soil conditions. The experimental data advance understanding and quantification of surface hydrologic and
erosion processes for the research domain and potentially for other patchy-vegetated rangeland landscapes elsewhere. Lastly, the unique nature of repeated measures spanning numerous treatments and time scales delivers a valuable dataset for examining long-term landscape vegetation, soil, hydrology, and erosion responses to various management actions, land use, and natural disturbances. The dataset is available from the National Agricultural Library at
https://data.nal.usda.gov/search/type/dataset (DOI: https://doi.org/10.15482/USDA.ADC/1504518; Pierson et al., 2019).





**Keywords:** ecohydrology; erosion; fire effects; infiltration; overland flow; prescribed fire; rainfall simulation; rangeland hydrology; runoff; sagebrush steppe; tree cutting; tree shredding; tree removal; woody plant encroachment

## 1 Introduction

Rangelands are one of the most common occurring sparsely-vegetated wildland landscapes around the world. These lands cover about half of the world's land surface and about 31% (> 300 million ha) of the land surface in the US (Havstad et al., 2009). The patchy vegetation structure typical to these water-limited landscapes regulates connectivity of runoff and erosion sources and processes and thus controls hillslope scale runoff and sediment transport (Pierson et al., 1994; Wainwright et al., 2000; Wilcox et al., 2003; Ludwig et al., 2005). Runoff and erosion in isolated bare patches on well-vegetated rangelands occur as splash-sheet (rainsplash and sheetflow) processes. Sediment entrained by raindrops and shallow sheetflow in bare patches typically moves a limited distance downslope before deposition immediately upslope of and within vegetated areas (Emmett, 1970; Reid et al., 1999; Puigdefábregas, 2005; Pierson and Williams, 2016). Disturbances such as intensive land use, plant community transitions, and wildfire can alter this resource-conserving vegetation structure and thereby facilitate increases in runoff and soil loss through enhanced connectivity of overland flow and sediment sources during rainfall events (Davenport et al., 1998; Wilcox et al., 2003; Pierson et al., 2011; Williams et al., 2014a, 2014b, 2018a). The negative ramifications of woody plant encroachment and wildfire have been extensively studied on rangelands around the World and have advanced understanding of runoff and erosion processes for these commonly occurring ecosystems (Schlesinger et al., 1990; Wainwright et al., 2000; Shakesby and Doerr, 2006; Shakesby, 2011; Pierson and Williams, 2016). Recent widespread plant community transitions and trends in wildfire activity and associated amplified runoff and erosion rates spanning rangelands to dry forests throughout the western US (Williams et al., 2014a) and elsewhere (Shakesby, 2011) underpin a need for compiling data sources that further contribute to process understanding and improved parametrization of rangeland hydrology and erosion predictive technologies.

Sagebrush rangelands in the western US are an extensive (> 500 000 km$^2$) and important vegetation type that have undergone substantial degradation associated with encroachment by pinyon (*Pinus* spp.) and juniper (*Juniperus* spp.) woodlands, invasions of fire-prone annual cheatgrass (*Bromus tectorum* L.), and altered fire regimes (Davies et al., 2011; Miller et al., 2011, 2019). Pinyon and juniper woodland encroachment of sagebrush vegetation can have negative hydrologic impacts (Miller et al., 2005; Petersen and Stringham, 2008; Pierson et al., 2007; Petersen et al., 2009; Pierson et al., 2010; Williams et al., 2014a, 2018a). Encroaching trees outcompete understory sagebrush and herbaceous vegetation over time and thereby increase bare ground and connectivity of runoff and sediment sources (Bates et al., 2000; Miller et al., 2000; Bates et al., 2005; Petersen et al., 2009; Pierson et al., 2010; Roundy et al., 2017). Extensive well-connected bare patches in the later stages of woodland encroachment propagate broad-scale runoff generation and soil loss during storms events. Runoff from splash-sheet processes during these events combine along hillslopes to form concentrated overland flow with high sediment detachment rates and ample transport capacity (Pierson et al., 2010; Williams et



al., 2014a, 2016a). Amplified soil loss over time perpetuates a woodland ecological state and long-term site degradation (Petersen et al., 2009). Land managers commonly employ various mechanical treatments and prescribed and natural fires to reduce tree cover and re-establish sagebrush vegetation and associated resource-conserving hydrologic function (Bates et al., 2000, 2005; Pierson et al., 2007; Bates et al., 2014; Miller et al., 2014; Roundy et al., 2014; Bates et al., 2017; Williams et al., 2018a). However, managers are challenged with predicting potential vegetation and ecohydrologic effects of tree removal across diverse woodland landscapes and with determining the appropriate type and timing of available treatment options. Invasions of fire-prone cheatgrass following prescribed and natural fires are particularly problematic. This annual grass commonly invades open patches on woodlands at lower elevations or on warmer sites, subsequently increases wildfire frequency, and potentially promotes long-term loss of surface soil and nutrients associated with recurrent burning and fire-induced runoff events (Pierson et al., 2011; Wilcox et al., 2012; Williams et al., 2014a).

Land managers around the World need improved understanding of runoff and erosion processes for the various disturbances common to rangelands and need improved tools for predicting responses to and making decisions on a host of management alternatives. Managers rely on local understanding and conceptual and quantitative science-based models to aid management decisions. Local knowledge is often limited and data necessary to populate conceptual and science-based models are likewise limited given vast rangeland domain. Vegetation and ground cover inventories and field-based experiments are primary resources for informing conceptual models (Petersen et al., 2009; Chambers et al., 2014; Williams et al., 2016a; Chambers et al., 2017). Rainfall simulation and overland flow experiments likewise provide data for developing, evaluating, and enhancing quantitative hydrology and erosion predictive technologies (Flanagan and Nearing, 1995; Robichaud et al., 2007; Wei et al., 2009; Nearing et al., 2011; Al-Hamdan et al., 2012a, 2012b, 2013, 2015, 2017; Hernandez et al., 2017). To this need, we present an ecohydrologic dataset containing 1021 experimental plots. The dataset consists of rainfall simulation (1300 plot runs, 0.5 $m^2$ to 13 $m^2$ scales) and overland flow (838 plot runs, ~9 $m^2$ scale) experimental data with paired measures of vegetation and ground cover, and surface soil physical properties spanning point to hillslope scales (Pierson et al., 2019). The experimental data were collected at multiple sagebrush rangelands in the Great Basin, USA, each with woodland encroachment and sampled in untreated conditions and following fire and mechanical tree-removal treatments over a 10 yr period. The dataset therefore represents diverse vegetation, ground surface, and surface soil conditions common to undisturbed and disturbed rangelands in the western US and elsewhere. The resulting dataset contributes to both process-based knowledge and provision of data for populating, evaluating, and improving conceptual and quantitative hydrology and erosion models.

## 2 Study Sites and Experimental Design

A series of vegetation, soils, rainfall simulation (Figures 1 and 2a-2c), and overland flow experiments (Figure 2d-2e) were completed at three pinyon and juniper woodlands historically vegetated as sagebrush shrublands. The study sites were selected from a network of sites as part of a larger study on the ecological impacts of invasive species and woodland encroachment into

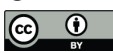



sagebrush ecosystems and the effects of sagebrush restoration practices, the Sagebrush Steppe
Treatment Evaluation Project (SageSTEP, www.sagestep.org). Study site climate, physical, and
vegetation attributes are provided in Table 1. The data were collected in years 2006-2015, with
sampling years varying by site and by treatment area within each site (see Table 2). Vegetation
and ground cover were patchy and sparse at the sites when the study began in 2006 (Table 1).
Tree-removal treatments (prescribed fire, tree cutting, tree shredding [bullhog]) were applied at
135 the Marking Corral and Onaqui sites in 2006 (late summer and autumn) to evaluate effectiveness
of pinyon and juniper removal in re-establishing sagebrush vegetation and ground cover,
improving hydrologic function, and reducing erosion rates. The Castlehead site burned by
wildfire in summer 2007 before tree-removal treatments could be applied, and, wildfire was
assessed as a prescribed natural-fire tree-removal treatment for that site. At all three sites, a cut-
140 tree (downed tree) treatment was placed across a subset of large-rainfall and overland-flow plot
bases (Figure 2d-2e) within the various treatments to measure effects of downed trees on surface
hydrology and erosion processes. This additional treatment was applied in 2007 and 2015 to
some plots in cut treatment areas at Marking Corral and Onaqui and in 2008 and 2009 in
unburned areas at Castlehead. Treatment applications and descriptions and the study
experimental design are explained in earlier papers by Pierson et al. (2010, 2013, 2014, 2015)
and by Williams et al. (2014a, 2018b, 2019a) and all treatments for each site each year are
provided in Table 2.

A suite of biological and physical attributes at each site were measured at point, small-
rainfall plot (0.5 m$^2$), overland-flow plot (~9 m$^2$), large-rainfall plot (13 m$^2$), and hillslope plot
(990 m$^2$) scales. Soil bulk density of the near-surface (0-5 cm depth) was sampled as a point
measure in interspace microsites between plants, shrub coppice microsites underneath shrub
canopies, and tree coppice microsites underneath three canopies. The bulk density sampling was
conducted by compliant cavity method within all treatment areas 1-2 yr after respective
treatments. Surface soil texture was quantified as a point measure using grab samples (0-2 cm
depth) from interspace, shrub coppice, and tree coppice microsites within all treatment areas at
Marking and Onaqui in 2006 prior to treatments and within unburned and burned treatment areas
at Castlehead in 2008. Vegetation and ground cover were measured at small-rainfall, large-
rainfall, and overland-flow plot scales and at the hillslope scale pre- and post-treatment in all
treatment areas at Marking Corral and Onaqui and in unburned and burned treatment areas at
160 Castlehead. Vegetation and ground cover measures at the hillslope scale (site characterization
plots) were conducted to describe site-level cover conditions prior to and over time after
treatment. Site characterization plots were installed and sampled prior to treatment (2006) in all
treatment areas at Marking Corral and Onaqui and were re-sampled 1 yr (2007) and 9 yr (2015)
after treatment. Castlehead site characterization plots were installed and sampled in unburned
and burned areas 1 yr after the fire (2008) and were re-sampled the 2$^{nd}$ year post-fire (2009).
Vegetation and ground cover measures on rainfall simulation and overland flow plots were used
to evaluate resisting and driving forces on surface hydrology and erosion processes and to
quantify treatment effects on cover components at those plot scales. Sampling of vegetation and
ground cover on rainfall simulation and overland flow plots in untreated areas (control and
170 unburned) and treated areas varied by site and year as described in Table 2.



Rainfall simulations and overland flow experiments were employed at the different plot scales to quantify specific scale-dependent runoff and erosion processes (Pierson et al., 2010; Williams et al., 2014a). Small-plot rainfall simulations (Figure 1) were applied to quantify runoff and erosion by splash-sheet processes. Each small rainfall plot was installed, as described by Pierson et al. (2010) and Williams et al. (2014a), to occur on either a tree coppice, shrub coppice, or interspace microsite (Figure 1b-1e). Small plots at Marking Corral and Onaqui were installed and sampled in control and all other treatment areas in 2006 before application of the tree-removal treatments and were left in place for subsequent sampling 1 yr (2007), 2 yr (2008), and 9 yr (2015) after treatment. Small plots at Castlehead were installed and sampled in unburned and burned areas 1 yr after the fire (2008) and left in place for subsequent sampling the 2$^{nd}$ year after fire (2009). Large-plot rainfall simulations (Figure 2a-2b) were used to quantify runoff and erosion from combined splash-sheet and concentrated overland flow processes. Each plot was installed, as described by Pierson et al. (2010) and Williams et al. (2014a), on either a tree zone (tree coppice and area just outside tree canopy drip line) or a shrub-interspace zone (intercanopy area between tree canopies) inclusive of shrub coppice and interspace microsites (Figure 2). Large plots at Marking Corral and Onaqui were installed and sampled in all treatment areas in 2006 immediately before treatment application (controls) and were extracted following sampling. New plots were installed and sampled in treatment areas at Marking Corral and Onaqui in 2007, 1 yr post-treatment, and were then extracted. Large rainfall plots at Castlehead were installed and sampled in unburned and burned areas in 2008, 1 yr after the fire, and were then extracted. Overland flow simulations (Figure 2d-de) were conducted on large rainfall plots (Figure 2a-2c) at Marking Corral and Onaqui in 2006 and 2007 immediately following respective rainfall simulations. Overland flow simulations were conducted in control and treated areas at those sites in 2008, but those plots were not subjected to rainfall simulation. Castlehead overland flow simulations in 2008, 1 yr post-fire, were run on large rainfall simulation plots following rainfall simulations and, in 2009, 2 yr post-fire, were run on newly installed plots without rainfall simulations. Overland flow experiments conducted on large-rainfall simulation plots had borders on all sides and contained a collection trough for runoff measurement at the plot base (Figure 2c; Pierson et al., 2010, 2013, 2015; Williams et al., 2014a). Overland flow simulations run independent of rainfall-simulation experiments were conducted on borderless plots, but contained a runoff collection trough at the downslope plot base (Figure 2d-2e; Pierson et al., 2013, 2015; Williams et al., 2014a, 2018b, 2019a).

## 3 Field Methods

### 3.1 Hillslope scale site characterization plots

Understory vegetation and ground cover and overstory tree cover at the hillslope scale at each site were sampled on 30 m × 33 m site characterization plots using a suite of line-point and belt transect methods and various tree measures (see Pierson et al., 2010; Williams et al., 2014a). Foliar and ground cover on each site characterization plot were recorded for 60 points (50 cm spacing) along each of five line-point transects (30 m in length; spaced 5-8 m apart) for a total of 300 sample points per plot. Percent cover by each sampled cover type was derived for each plot





as the number of respective cover type hits divided by the total number of points sampled. Multiple canopy layers were possible and therefore the total foliar cover across all sampled cover types potentially exceeded 100%. The number of live tree seedlings 5-50 cm height and shrubs exceeding 5-cm height were quantified along three belt transects on each plot. Each of the three belt transects on each plot were centered along a foliar/ground cover line-point transect, sized 2 m wide × 30 m long, and spaced 6 m apart. Shrub and tree seedling densities were calculated for

each plot as the total number of respective individuals tallied along the three belt transects divided by total belt transect area (180 m$^2$). The number of live trees > 0.5 m in height was quantified for each plot, and tree height and minimum and maximum crown diameters were measured for each live tree. A crown radius for each live tree was derived as one-half the average of measured minimum and maximum crown diameters. Individual tree crown area (tree

cover) was calculated as equivalent to the area of a circle, derived with the respective crown radius. Total tree cover for each plot was quantified as the sum of measured tree cover values on the plot.

### 3.2 Small-rainfall simulation plots and experiments

Foliar cover, ground cover, and ground surface roughness on all small-rainfall plots were quantified using point frame methods explained in Pierson et al. (2010). Foliar and ground cover on each plot were sampled at 15 points spaced 5 cm apart along each of seven transects spaced 10 cm apart and oriented parallel to hillslope contour (105 sample points per plot). Percent cover

for each cover type sampled on each plot was derived from the frequency of respective cover type hits divided by the total number of points sampled. Multiple canopy layers were allowed and therefore total foliar cover across all cover types potentially exceeded 100%. A relative ground surface height at each sample point on each plot was determined by metal ruler as the distance between the ground surface and a level-line (top of point frame). Ground surface

roughness for each plot was then derived as the mean of standard deviations of ground surface heights for each of the transects sampled on the respective plot. Litter depth on each plot was measured along the outside edge of the two plot borders located perpendicular to the hillslope contour. Measurements were made to the nearest 1 mm using a metal ruler at four evenly spaced points (15-cm apart) along the two plot borders. An average litter depth was derived for each plot

as the averaged of the eight litter depth measures.
        Soil water repellency of the mineral soil surface and at depths near the mineral soil surface (0-5 cm depths) was measured immediately adjacent (~ 50 cm away) to each small-rainfall plot immediately before rainfall simulation using the water drop penetration time (WDPT) method (see Pierson et al., 2010). Litter and ash cover were carefully removed from the

mineral soil surface prior to application of the WDPT. Eight water drops (~ 3-cm spacing) were then placed on the mineral soil surface and the time required for infiltration of each drop was recorded up to a 300-s maximum. The WDPT was then repeated at 1-cm soil depth increments until 5-cm soil depth was reached. For each sampled depth, 1 cm of soil was excavated immediately underneath the previously sampled area and the WDPT procedure was repeated

with eight drops. A mean WDPT for each sampled soil depth on each plot was recorded as the average of the eight WDPT (s) samples at the respective depth. Soils were classified as wettable



where mean WDPT < 5 s, slightly water repellent where mean WDPT ranged 5 s to 60 s, and strongly water repellent where mean WDPT > 60 s.

Surface soil moisture and aggregate stability were also sampled for each small-rainfall plot prior to rainfall simulations. Soil samples were collected at 0-5 cm depth immediately adjacent to each small rainfall plot and were subsequently analyzed in the laboratory for gravimetric soil water content. Some samples were excluded from the dataset due to poor sealing of soil cans in the field. Aggregate stability of the surface soil on each plot was determined using a modified sieve test on six soil peds approximately 2-3 mm thick and 6-8 mm in diameter (see

Pierson et al., 2010). Each soil ped sampled on each plot was assigned to one of the following classes, as defined by Herrick et al. (2005): (1) > 10% stable aggregates, 50% structural integrity lost within 5 s, (2) > 10% stable aggregates, 50% structural integrity lost within 5-30 s, (3) > 10% stable aggregates, 50% structural integrity lost within 30-300 s, (4) 10-25% stable aggregates, (5) 25-75% stable aggregates, or (6) 75-100% stable aggregates. An average

aggregate stability was derived for each plot as the arithmetic average of the classes assigned to the six aggregate samples for the respective plot.

        Rainfall was applied to small-rainfall plots at approximate intensities of 64 mm h$^{-1}$ (dry run) and 102 mm h$^{-1}$ (wet run) for 45 min as explained in Pierson et al. (2010). The dry run was applied to dry antecedent soil conditions, and the wet run was applied to wet soil conditions, ~

30 min after the dry run. Rainfall was applied to small-rainfall plots by a Meyer and Harmon-type portable oscillating-arm rainfall simulator with 80-100 Veejet nozzles (Figure 1a; Meyer and Harmon, 1979; Pierson et al. 2010, 2013, 2014; Williams et al., 2014a, 2018b, 2019a). The applied rainfall kinetic energy (200 kJ ha$^{-1}$ mm$^{-1}$) and raindrop size (2 mm) were within approximately 70 kJ ha$^{-1}$ mm$^{-1}$ and 1 mm respectively of values reported for natural convective

rainfall (Meyer and Harmon, 1979). Rainfall amount applied to each plot during rainfall simulation was estimated by integrating a pan catch of a 5-min calibration run prior to each rainfall simulation plot run. Total rainfall amount was estimated on plots where debris and/or vegetation prevented placement of calibration pans. In such cases, the estimated rainfall amount was derived as the average of all calibration runs for the respective simulation date. Timed plot

runoff samples were collected at 1-3-min intervals throughout each 45-min rainfall simulation and were subsequently analyzed in the laboratory for runoff volume and sediment concentration. Cumulative runoff and sediment amounts were obtained for each runoff sample by weighing the sample before and after drying at 105°C. A mean runoff rate (mm h$^{-1}$ and L min$^{-1}$) was derived for each sample interval as the interval runoff divided by the interval time. Sediment discharge (g

s$^{-1}$) for each sample interval was calculated as the cumulative sediment for the sample interval divided by the interval time. Sediment concentration for each sample interval was obtained by dividing cumulative sediment by cumulative runoff (g L$^{-1}$). Some field samples were discarded from the final dataset because of laboratory errors or various issues noted on field datasheets (i.e., spillage, bottle overrun, etc.).

## 3.3 Large-rainfall simulation plots and experiments

Vegetation and ground cover were measured on large-rainfall simulation plots using line-point methods as described by Pierson et al. (2010) and Williams et al. (2014a). Foliar cover and





ground cover on large-rainfall plots were recorded for 59 points with 10-cm spacing along each of five transects (6 m long, spaced 40 cm apart) oriented perpendicular to the hillslope contour, 295 sample points per plot. The percentage cover by each sampled cover type for each plot was derived as the number of point contacts or hits for each respective life form divided by the total number of points sampled on the respective plot. Multiple canopy layers were allowed and

therefore total foliar cover across all sampled cover types potentially exceeded 100%. Cut trees placed on a subset of rainfall simulation plots (see experimental design above) were excluded from foliar and ground cover measurements. However, various attributes of downed trees (i.e., length [height], crown width, etc.) were measured and are reported. Ground surface roughness for each plot was calculated as the average of the standard deviations of ground surface heights

measured across the line-point cover transects. The relative ground-surface height at each sample point was calculated as the distance between a survey transit level-line above the point and the ground surface. Distances in excess of 20 cm between plant canopies (canopy gaps) and plant bases (basal gaps) were measured along each of the line-point transects on each plot. Average canopy and basal gap sizes were calculated for each plot as the mean of all respective gaps

measured in excess of 20 cm. Additionally, maximum canopy and basal gap sizes were calculated for each plot as the maximum of all respective gaps measured in excess of 20 cm. Percentages of canopy gaps and basal gaps representing 50-cm incremental gap classes (e.g., 51-100 cm, 101-150 cm, etc.) were derived for each transect and averaged across the transects on each plot to determine gap-class plot means.

Rainfall was applied to pairs of large-rainfall plots (Figure 2a-2b) at the same dry-run and wet-run target rates and sequence and durations as described above for small-rainfall plots (Pierson et al., 2010; Williams et al., 2014a). Each paired-rainfall simulation was run with a Colorado State University (CSU) type rainfall simulator (Figure 2a-2b; Holland, 1969). The CSU-type design delivers rainfall energy at approximately 70% of that for a natural convective

rainfall event and produces rainfall drop diameters within approximately 1 mm of natural rainfall (Holland, 1969; Neff, 1979). The applied simulator design consists of seven stationary sprinklers evenly spaced along each of the outermost borders of the respective rainfall-plot pair, with each sprinkler elevated 3.05 m above the ground surface. Total rainfall applied to large-rainfall plots was quantified from the average of six plastic rainfall depth gages organized in a

uniform grid within each plot. Runoff from direct rainfall on the large-plot collection troughs (trough catch, Figure 2b) was quantified by sampling collection trough runoff before plot-generated runoff occurred. Once plot runoff occurred, timed samples of runoff were collected at 1-3-min intervals throughout each 45-min simulation run and were subsequently analyzed in the laboratory for runoff volume and sediment concentration as with small-plot rainfall simulation

samples. Sample weights were adjusted to appropriately account for trough catch, as described by Pierson et al., 2010. Some field samples were discarded from the final dataset because of laboratory errors or various issues noted on field datasheets (i.e., spillage, bottle overrun, etc.). Runoff and erosion rates were determined consistent with methods for small-plot rainfall simulations.

### 3.4 Overland-flow simulation plots and experiments

Vegetation and ground cover on overland-flow plots were measured using methods consistent with those on large-rainfall simulation plots. For overland-flow plots that underwent rainfall simulation, foliar and ground cover measures were derived from the large-rainfall plot line-point transect data, but were restricted to the lower 4 m of the respective plots. Foliar and ground cover on overland-flow plots not subjected to rainfall simulations were recorded at 24 points with 20-
350 cm spacing, along each of nine line-point transects (4.6 m in length, spaced 20 cm apart) oriented perpendicular to the hillslope contour, for a total of 216 points per plot. Percentage cover for each cover type sampled on each plot was derived from the number of point contacts or hits for each respective cover type divided by the total number of points sampled within the plot. As on large-rainfall plots, total foliar cover across all cover types potentially exceeded 100% given
multiple canopy were allowed. Cut trees placed on a subset of overland-flow plots (see experimental design above) were excluded from foliar and ground cover measurements. However, various attributes of downed trees (i.e., length [height], crown width, etc.) were measured and are reported. The ground surface roughness for each overland-flow plot was calculated as the average of the standard deviations of the ground surface heights across the
foliar/ground cover line-point transects. The relative ground-surface height at each cover sample point was calculated as the distance between a survey transit level line above the respective sample point and the ground surface. Canopy and basal gaps exceeding 20 cm on overland-flow plots were recorded along each line-point transect. Average and maximum canopy and basal gaps were derived consistent with methods for large-rainfall simulation plots. Percentages of
canopy and basal gaps representing 50-cm incremental gap classes (e.g., 51-100 cm, 101-150 cm, etc.) were derived for each transect and averaged across the transects on each plot to determine gap-class plot means, similar as on large-rainfall plots.

Datalogger-controlled flow regulators (see Pierson et al., 2010, 2013, 2015; Williams et al., 2014a, 2018b, 2019a) were used to apply concentrated flow release rates of 15, 30, and 45 L
370 min$^{-1}$ to each overland-flow plot. Flow was routed into and through a metal box filled with Styrofoam pellets and was released through a 10-cm wide mesh-screened opening at the box base (Figure 2d; see Pierson et al., 2010). Each flow release on each plot was applied for 12 min from a single release-point located 4 m upslope of the collection trough apex. Flow release rate progression on each plot was consecutive from 15 L min$^{-1}$ to 30 L min$^{-1}$ to 45 L min$^{-1}$. Flow
samples were collected at various time intervals (usually 1-min to 2-min) for each 12-min simulation at each release rate. As with rainfall simulation samples, runoff samples were taken to the laboratory, weighed, oven-dried at 105°C, and then re-weighed to determine the runoff rate and sediment concentration. Also as noted above for rainfall simulation samples, a small number of runoff samples were discarded because of laboratory errors or various issues noted on field
datasheets (i.e., spillage, bottle overrun, etc.). Runoff and sediment variables for each flow release rate were calculated for an 8-min time period starting at runoff initiation. The resulting 8-min runoff and sediment variables were derived as explained for the 45-min rainfall simulations. The velocity of overland flow was measured using a concentrated salt tracer applied into the flow and electrical conductivity probes to track the mean transit time of the tracer over a set flow
path length (usually 3 m; Pierson et al., 2010, 2013, 2015; Williams et al., 2014a, 2018b, 2019a).





The width, depth, and a total rill area width (TRAW) of overland flow were measured along flow cross-sections 1 m, 2 m, and 3 m downslope from the flow release point. The TRAW variable represents the total width between the outermost edges of the outermost flow paths at the respective cross section (Pierson et al., 2010). Overland flow simulations conducted on large-
rainfall simulation plots at Marking Corral and Onaqui in 2006 and 2007 and at Castlehead in 2008 were run approximately two hours after respective rainfall simulations. Overland flow simulations on plots not subjected to rainfall simulation at Marking Corral and Onaqui in 2008 and 2015 and at Castlehead in 2008 were conducted on soils pre-wet with a gently misting sprinkler (see Pierson et al., 2013, 2015; Williams et al., 2014, 2018, 2019a).

## 4 Data Application

Subsets of the dataset have been used to improve understanding of rangeland hydrologic and erosion processes, assess the ecohydrologic impacts of wildland fire and management practices
on sagebrush rangelands, and improve and enhance rangeland hydrology and erosion models. Examples of data use for such applications are presented in Figures 3-5. Pierson et al. (2010) applied pre-treatment data across all plot-scales and experiment types from Marking Corral and Onaqui to evaluate the ecohydrologic impacts of woodland encroachment on sagebrush rangelands. Studies by Pierson et al. (2014, 2015) assessed the initial (1st and 2nd year) effects of
prescribed fire and mechanical tree removal treatments on vegetation, ground cover, and hydrology and erosion processes at Marking Corral and Onaqui. Williams et al. (2014a) applied vegetation, ground cover, rainfall simulation and overland flow experiments from unburned and burned areas at Castlehead to evaluate the utility of fire to reverse the negative ecohydrologic impacts of juniper encroachment on rangelands and to frame conceptual concepts on process
connectivity for burned and degraded rangelands (Figure 4). Pierson et al. (2013 and 2015) evaluated the immediate effects of cut-downed trees on runoff and erosion processes on woodlands. Williams et al. (2018b, 2019a, 2019b) applied data from all experimental plot scales and methods in untreated and treated areas at Marking Corral and Onaqui to evaluate the long-term ecohydrologic impacts of prescribed fire and mechanical tree-removal treatments on
woodland-encroached sagebrush steppe (Figure 5). Al-Hamdan et al. (2012a, 2012b, 2013, 2015, 2017) applied subsets of the data to develop, test, and enhance various parameter estimation equations for flow hydraulics and erodibility parameters in the Rangeland Hydrology and Erosion Model (RHEM). Collectively, these studies have improved understanding of rangeland hydrology and erosion processes and informed both conceptual and quantitative models
applicable to assessment and management of diverse rangelands (McIver et al., 2014; Pierson and Williams, 2016; Williams et al., 2016a, 2016b, 2016c; Hernandez et al., 2017; Williams et al., 2018a).

## 5 Data Availability

The full dataset is available from the National Agricultural Library website at https://data.nal.usda.gov/search/type/dataset (DOI: https://doi.org/10.15482/USDA.ADC/1504518; Pierson et al., 2019). The suite of files therein





includes an abbreviated description and field methods; a data dictionary; geographic information
for study sites; photographs of the study sites, field experiments, and experimental plots; and
datafiles for vegetation, ground cover, soils, and hydrology and erosion time series measures
spanning the associated plots scales. Subset examples of the datafiles are shown in Tables 4 (site
level soil particle size and bulk density), 5 (site characterization plots), 6 (small-rainfall plot
attributes), 7 (large-rainfall plot attributes), 8 (overland-flow plot attributes), 9 (small-plot
rainfall simulation time series), 10 (large-plot rainfall simulation time series), and 11 (overland-
flow simulation time series).

## 6 Summary and Conclusions

Rangelands are uniquely managed using ecological principles. As such, our functional
understanding of regulating ecohydrologic processes, such as soil conservation and runoff
moderation, are limited by our ability to track these processes in the context of interdependent
land management decisions. Pinyon-juniper encroachment into sagebrush shrublands and the
resulting management actions provide a model system for observing hydrologic processes under
disturbances and interventions typical of extensively managed rangelands. To provide detailed
understanding of ecohydrologic processes under realistic management conditions, we collected
long-term data at multiple sites, spatial scales, and treatments. The combined dataset includes
1021 experimental plots and contains vegetation, ground cover, soils, hydrology, and erosion
data spanning multiple spatial scales and diverse vegetation, ground cover, and surface soil
conditions from three study sites and five different study years. The dataset includes 57 plots
from the hillslope scale (site characterization plots), 528 small rainfall simulation plots, 146 large
rainfall simulation plots, and 290 overland-flow simulation plots. The hydrology and erosion
experiments provide time series data for small-rainfall plot, large-rainfall plot, and overland-flow
plot simulations. After excluding some time series rainfall- and overland-flow simulation data
due to various lab and equipment failures, the final time series dataset contains 1020 small-
rainfall, 280 large-rainfall, and 838 overland-flow plot-run hydrographs and sedigraphs if plots
without runoff are retained. Retaining only plots that generated runoff results in a time series
dataset of 749 small-rainfall, 251 large-rainfall, and 719 overland-flow plot simulation
hydrographs and sedigraphs. Overall, the hydrology and erosion time series dataset totals to 2138
hydrographs/sedigraphs including plots with no runoff and 1719 hydrographs/sedigraphs for
plots that generated runoff. The methodology employed and resulting experimental data improve
understanding of and provide quantification of separate scale-dependent (e.g., rainsplash and
sheetflow) and combined (e.g., interrill and concentrated flow/rill) surface hydrology and erosion
processes for sagebrush rangelands and pinyon and juniper woodlands in the Great Basin before
and after tree removal and for sparsely vegetated sites elsewhere. This separate and combined
experimental approach yields a valuable data source for testing and improving isolated process
parameterizations in quantitative hydrology and erosion models. The long-term nature of the
dataset is unique and provides a substantial database for populating conceptual ecological models
of changes in vegetation, ground cover conditions, and soils resulting from management
practices and disturbances. Likewise, the combined data on short-term and long-term



ecohydrologic impacts of management practices and fire provide valuable insight on trends in ecohydrologic recovery of rangeland ecosystems.

**Author contributions.** Frederick B. Pierson, C. Jason Williams, Patrick R. Kormos, and Osama
Z. Al-Hamdan participated in the experimental design, data collection and reduction, and compilation of the dataset and manuscript. Justin C. Johnson contributed to data reduction and compilation of the dataset and manuscript. All authors contributed to revisions of the submitted manuscript.

**Competing interests.** The authors declare that they have no conflict of interest.

**Acknowledgements.** This paper is contribution number 135 of the Sagebrush Steppe Treatment Evaluation Project (SageSTEP, www.sagestep.org), funded by the US Joint Fire Science Program, US Department of Interior (USDI) Bureau of Land Management, and US National
Interagency Fire Center. Additional funding provided by the US Department of Agriculture (USDA) Agricultural Research Service (ARS). The authors thank the USDI Bureau of Land Management and the USDA Forest Service for implementation of the land management treatments and site access in collaboration with the SageSTEP study. We are also grateful for land access and infrastructural support provided by Mike and Jeannie Stanford during our field
experiments at the Castlehead site. We thank Barry Caldwell and Zane Cram of the USDA-ARS Northwest Watershed Research Center, Boise, ID, USA, for field support throughout the study. We likewise thank Steve Van Vactor of the USDA-ARS Northwest Watershed Research Center for database support. We are grateful for field supervision of data collection and laboratory work provided by Jaime Calderon, Matthew Frisby, Kyle Lindsay, and Samantha Vega over various
495     years of the research study. We thank Dr. Ben Rau and the Desert Research Institute, Reno, Nevada, USA, for assistance with processing soil samples. The USDA is an equal opportunity provider and employer. Mention of a proprietary product does not constitute endorsement by USDA and does not imply its approval to the exclusion of the other products that may also be suitable.

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

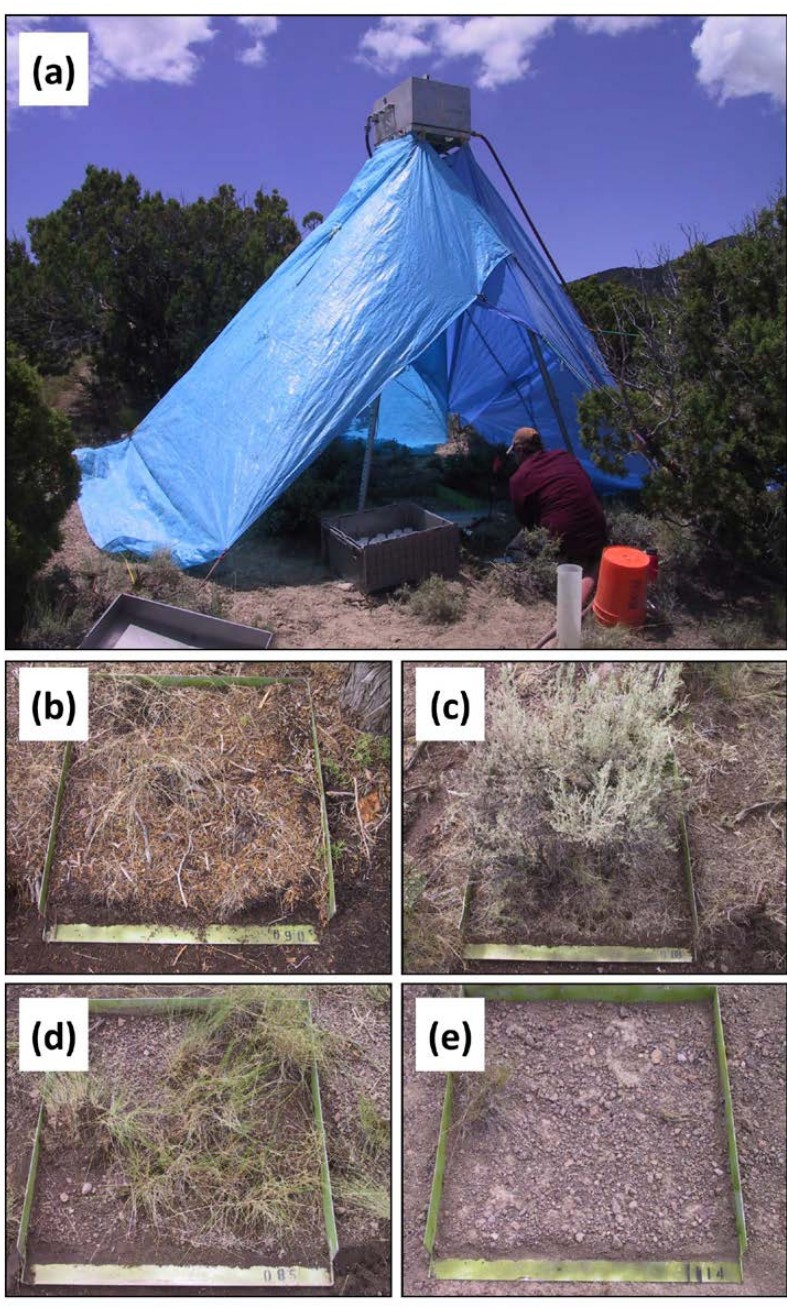

**Figure 1.** Photographs of small-plot rainfall simulator (**a**) and example small-rainfall plots on tree coppice (**b**), shrub coppice (c), and interspace (**d** and **e**) microsites as applied in this study.

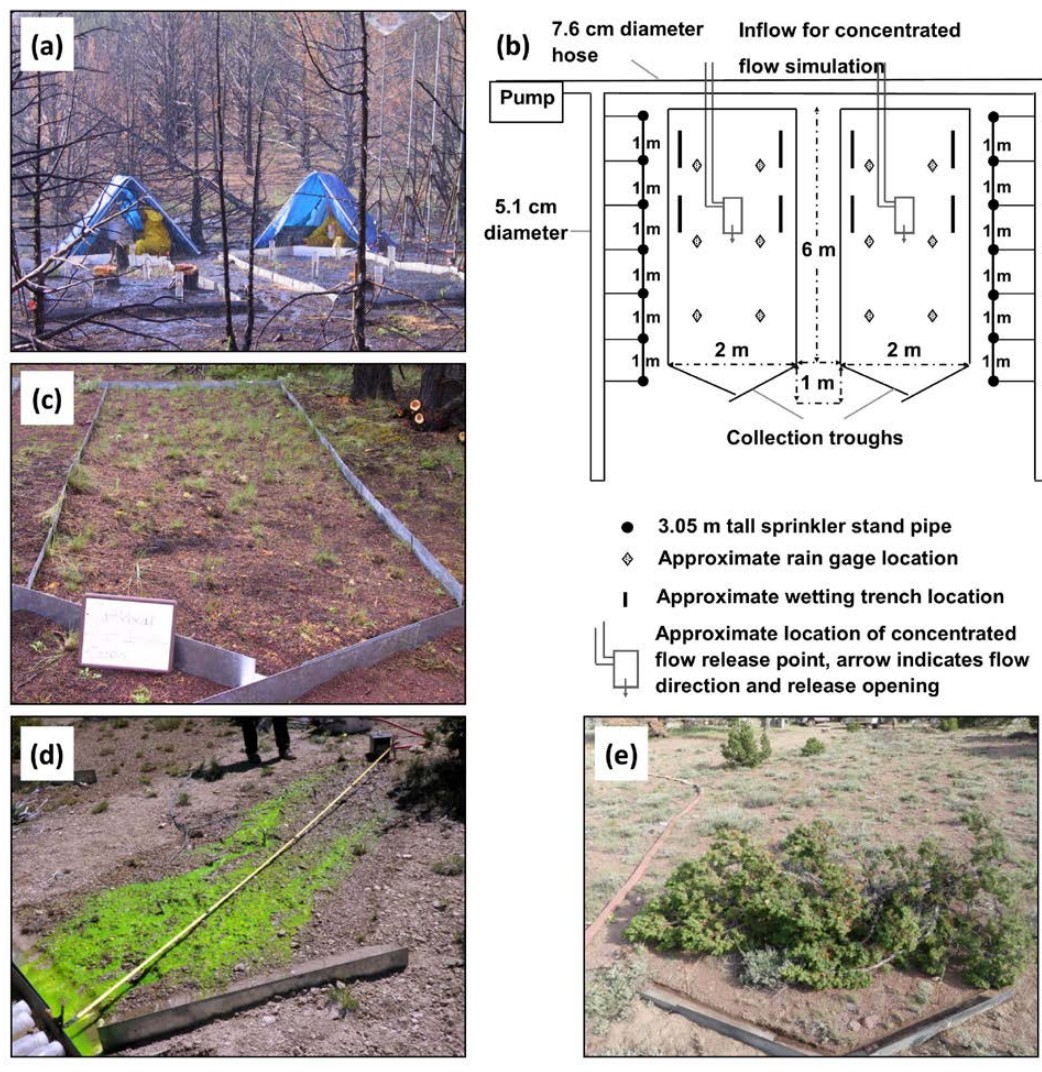

**Figure 2.** Images showing paired large-rainfall plots during rainfall simulations (**a**), experimental set-up of paired large-rainfall plot simulation experiments (**b**), a fully-bordered large-rainfall simulation plot on a tree coppice microsite (**c**), a borderless overland-flow simulation plot and experiment on an intercanopy (shrub-interspace) microsite (**d**), and a borderless overland-flow simulation plot with a cut, downed tree on an intercanopy microsite, all as respective examples as applied in this study.

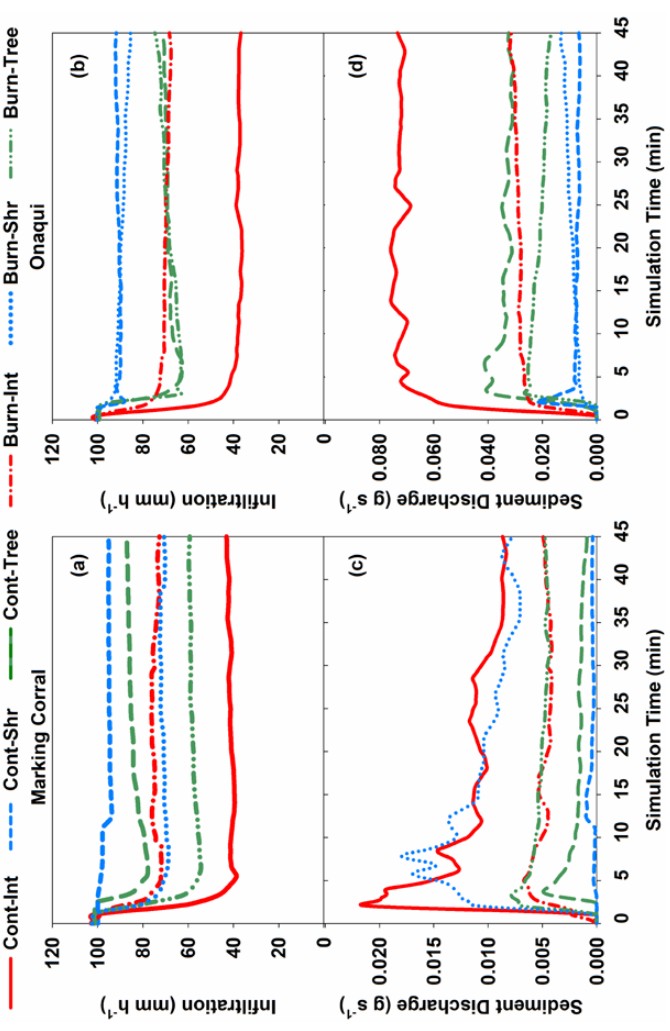

**Figure 3.** Example infiltration (**a** and **b**), calculated as applied rainfall minus measured runoff, and sediment discharge (**c** and **d**) time series data generated from a subset of the small-plot rainfall simulation dataset. Example sub-dataset is from wet-run rainfall simulations in untreated (Cont) and burned (Burn) interspace (Int), shrub coppice (Shr), and tree coppice (Tree) microsites at the Marking Corral and Onaqui study sites 9 yr following prescribed fire. The data illustrate the long-term impacts of burning and associated changes in surface conditions on infiltration and sediment discharge. Figure modified from Williams et al. (2018b).

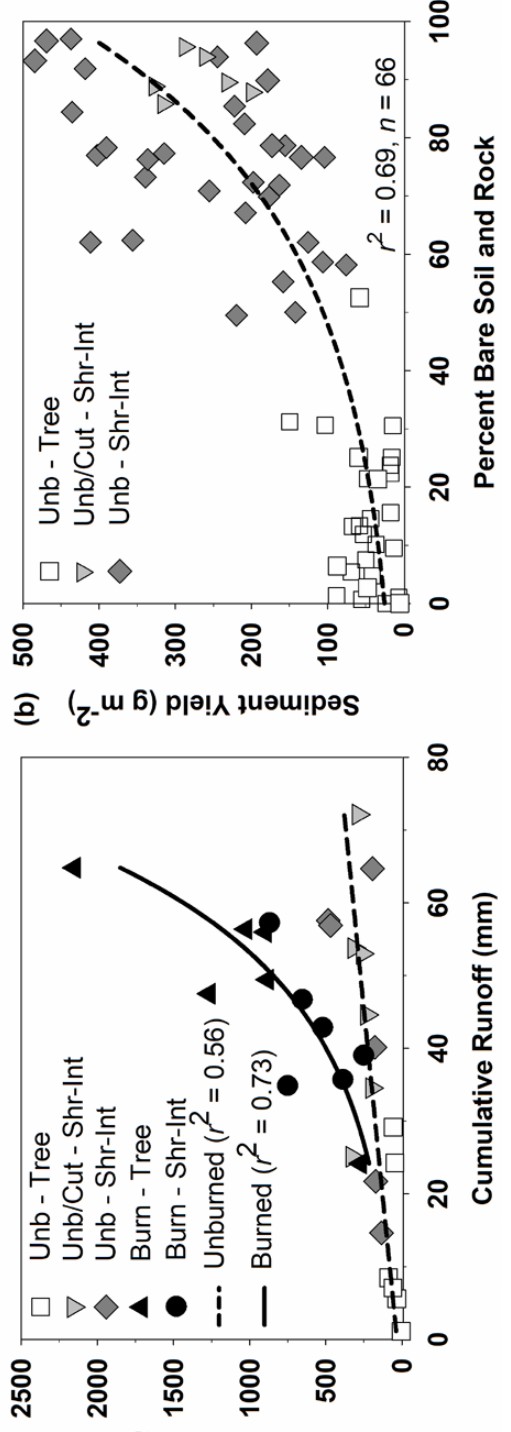

**Figure 4.** Example relationships/correlations in large-rainfall plot cumulative runoff and sediment yield for unburned (untreated [Unb] and cut [Cut] treatments) and burned (Burn) tree (Tree) and intercanopy (shrub-interspace, Shr-Int) plots at the Castlehead site (**a**) and bare ground (bare soil plus rock cover) and sediment yield for unburned (Unb) and cut treatment (Cut) tree and intercanopy plots across all study sites (Castlehead, Marking Corral, and Onaqui) (**b**). The relationship in runoff and sediment yield (**a**) demonstrates the initial (1 yr) impact of burning on sediment availability and elevated sediment delivery (for tree coppices in this study) as commonly reported in fire studies (Pierson and Williams, 2016). The relationship in bare ground and sediment yield (**b**) shows the typical increase in sediment yield where bare ground exceeds 50-60% as commonly reported for rangelands (Pierson et al., 2008, 2009; Williams et al., 2014a). Figures modified from Pierson et al. (2013) and Williams et al. (2014a).

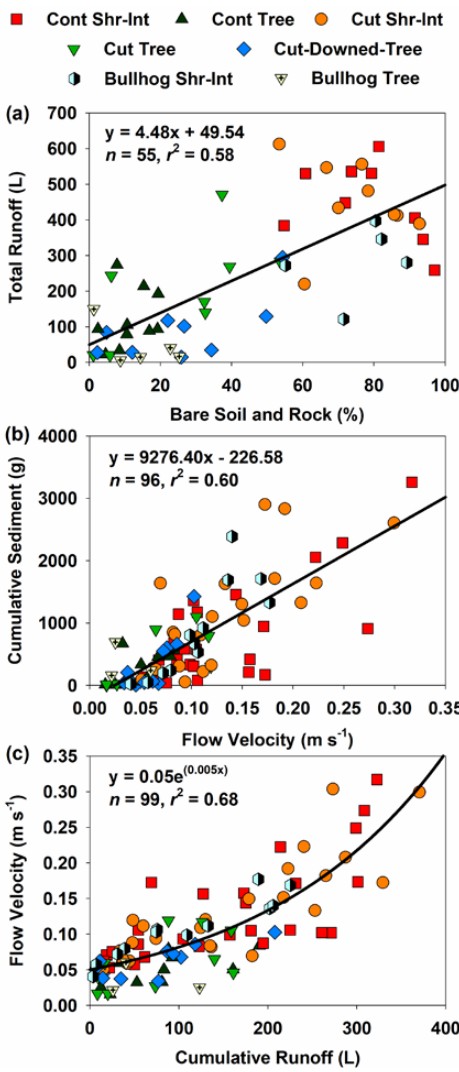

2
**Figure 5.** Example relationships/correlations in runoff and bare ground (bare soil plus rock
cover; **a**), cumulative sediment and overland flow velocity (**b**), and overland flow velocity and
runoff (**c**) derived from a subset of the overland flow dataset for Marking Corral and Onaqui
sites, as presented in Williams et al. (2019a). Data from overland flow simulations on
untreated/control (Cont) plots, cut treatment (Cut) plots without and with a cut, downed tree
(Cut-Downed Tree), and bullhog plots (Bullhog, Onaqui site only) in tree (Tree) and intercanopy
(shrub-interspace, Shr-Int) microsites 9 yr after respective tree removal treatments. The data
demonstrate that, for the studied conditions, runoff is largely regulated by bare ground, sediment
delivery is controlled by flow velocity, and flow velocity is strongly correlated with the amount
or runoff.

**Table 1** Topography, climate, soil, tree cover, and understory vegetation at the Castlehead, Marking Corral, and Onaqui sites prior to treatments. Data from Pierson et al. (2010, 2015) or Williams et al. (2014a) except where indicated by footnote.

| | Castlehead, Idaho, USA | Marking Corral, Nevada, USA | Onaqui, Utah, USA |
|---|---|---|---|
| Woodland community | western juniper[1] | single-leaf pinyon[2]/Utah juniper[3] | Utah juniper[3] |
| Elevation (m) - Aspect | 1750 – SE facing | 2250 – W to SW facing | 1720 – N to NE facing |
| Mean annual precip. (mm) | 364[4] | 299[4] | 298[4] |
| Mean annual air temp. (°C) | 7.4[4] | 6.9[4] | 9.2[4] |
| Slope (%) | 10-25 | 10-15 | 10-15 |
| Parent rock | basalt and welded tuff[5] | andesite and rhyolite[6] | sandstone and limestone[7] |
| Soil association | Mulshoe-Squawcreek-Gaib[5] | Segura-Upatad-Cropper[6] | Borvant[7] |
| Depth to bedrock (m) | 0.5-1.0[5] | 0.4-0.5[6] | 1.0-1.5[7] |
| Soil surface texture | sandy loam, 59% sand, 37% silt, 4% clay | sandy loam, 66% sand, 30% silt, 4% clay | sandy loam, 57% sand, 37% silt, 7% clay |
| Tree canopy cover (%)[8] | 26[1] | 15[2], 10[3] | 26[3] |
| Trees per hectare[8] | 158[1] | 329[2], 150[3] | 476[3] |
| Mean tree height (m)[8] | 5.2[1] | 2.3[2], 2.4[3] | 2.4[3] |
| Juvenile trees per hectare[9] | 28[1] | 296[2], 139[3] | 154[3] |
| Shrubs per hectare[10] | 2981 | 12065 | 4914 |
| Intercanopy bare ground (%)[11] | 88 | 64 | 79 |
| Common understory plants | *Artemisia tridentata* Nutt. ssp. *wyomingensis* Beetle & Young; *Artemisia nova* A. Nelson; *Artemisia tridentata* Nutt. ssp. *vaseyana* (Rydb.) Beetle; *Purshia* spp.; *Poa secunda* J. Presl; *Pseudoroegneria spicata* (Pursh) A. Löve; *Festuca idahoensis* Elmer; and various forbs | | |

[1] *Juniperus occidentalis* Hook.
[2] *Pinus monophylla* Torr. & Frém.
[3] *Juniperus osteosperma* [Torr.] Little.
[4] Estimated from 4 km grid for years 1989-2018 from Prism Climate Group (2019).
[5] Natural Resources Conservation Service (NRCS) (2003).
[6] NRCS (2007).
[7] NRCS (2006).
[8] Trees ≥ 50 cm height, for Castlehead includes data from Williams et al. (2014a) and one additional year.
[9] Trees 5 to 50 cm height, for Castlehead mean based on data from Williams et al. (2014a) and one additional year.
[10] Shrubs ≥ 5 cm height, for Castlehead mean based on data from Williams et al. (2014a) and one additional year.
[11] Area between tree canopies consisting of shrubs, grasses, and interspaces between plants (shrub-interspace zone).





**Table 2.** Number of plots sampled by plot type (site characterization and small plot rainfall, large plot rainfall, and overland flow simulations) at each study site (Castlehead, Marking Corral, and Onaqui) by treatment and microsite (small plots - tree coppice, shrub coppice, and interspace; large plots and overland flow – tree zone and shrub-interspace zone [intercanopy]) combination each year of the study. Control refers to untreated areas at Marking Corral and Onaqui sites. Unburned refers to areas immediately adjacent to, but outside the wildfire area (burned treatment) at the Castlehead site. Downed tree sub-treatments (cut-downed tree and unburned-downed tree) refer to plots with a single downed tree across each respective plot within the specified associated treatment (cut or unburned). Tree and shrub coppice microsites are areas underneath or previously (prior to treatment) underneath tree and shrub canopy, respectively. Interspace microsites are areas between tree and shrub coppice microsites. Tree zone microsites are areas underneath, or previously underneath, and immediately adjacent (just outside canopy drip line) to a tree canopy. Shrub-interspace zones are the areas between tree canopies, collectively inclusive of shrub coppice and interspace microsites [the intercanopy].

| Year | Treatment | Castlehead | Marking Corral | Onaqui |
|---|---|---|---|---|
| 2006 | Control | - | 6 | 9 |
| 2007 | Bullhog | - | - | 3 |
| | Burned | - | 3 | 3 |
| | Cut | - | 3 | 3 |
| 2008 | Unburned | 3 | - | - |
| | Burned | 3 | - | - |
| 2009 | Unburned | 3 | - | - |
| | Burned | 3 | - | - |
| 2015 | Bullhog | - | - | 3 |
| | Burned | - | 3 | 3 |
| | Cut | - | 3 | 3 |

| | | Castlehead | | | Marking Corral | | | Onaqui | | |
|---|---|---|---|---|---|---|---|---|---|---|
| Year | Treatment | Tree Coppice | Shrub Coppice | Interspace | Tree Coppice | Shrub Coppice | Interspace | Tree Coppice | Shrub Coppice | Interspace |
| 2006 | Control | - | - | - | 24 | 13 | 23 | 23 | 21 | 36 |
| 2007 | Control | - | - | - | 7 | 5 | 8 | 4 | 3 | 3 |
| | Bullhog | - | - | - | - | - | - | 10 | 10 | 30 |
| | Burn | - | - | - | 8 | 4 | 8 | 5 | 5 | 10 |
| 2008 | Control/ Unburned | 8 | 8 | 8 | 4 | 2 | 4 | 4 | 3 | 3 |
| | Burned | 5 | 5 | 10 | 8 | 4 | 8 | 5 | 5 | 10 |
| 2009 | Unburned | 3 | 3 | 4 | - | - | - | - | - | - |
| | Burned | 5 | 5 | 10 | - | - | - | - | - | - |
| 2015 | Control | - | - | - | 8 | 4 | 6 | 8 | 6 | 6 |
| | Bullhog | - | - | - | - | - | - | 5 | 5 | 10 |
| | Burned | - | - | - | 8 | 4 | 6 | 5 | 5 | 10 |
| | Cut | - | - | - | 8 | 4 | 6 | 5 | 5 | 10 |

| | | Castlehead | | Marking Corral | | Onaqui | |
|---|---|---|---|---|---|---|---|
| Year | Treatment | Tree Zone | Shrub-Interspace Zone | Tree Zone | Shrub-Interspace Zone | Tree Zone | Shrub-Interspace Zone |
| 2006 | Control | - | - | 12 | 12 | 18 | 18 |
| 2007 | Bullhog | - | - | - | - | 4 | 4 |
| | Burned | - | - | 6 | 6 | 6 | 6 |
| | Cut | - | - | - | 6 | - | 6 |
| | Cut-Downed Tree | - | - | - | 6 | - | 6 |
| 2008 | Unburned | 6 | 6 | - | - | - | - |
| | Unburned-Downed Tree | - | 6 | - | - | - | - |
| | Burned | 6 | 6 | - | - | - | - |

| | | Castlehead | | Marking Corral | | Onaqui | |
|---|---|---|---|---|---|---|---|
| Year | Treatment | Tree Zone | Shrub-Interspace Zone | Tree Zone | Shrub-Interspace Zone | Tree Zone | Shrub-Interspace Zone |
| 2006 | Control | - | - | 12 | 12 | 18 | 18 |
| 2007 | Bullhog | - | - | - | - | 4 | 4 |
| | Burned | - | - | 6 | 6 | 6 | 6 |
| | Cut | - | - | - | 6 | - | 6 |
| | Cut-Downed Tree | - | - | - | 6 | - | 6 |
| 2008 | Control Unburned | 6 | 6 | 3 | 3 | 2 | 2 |
| | Unburned-Downed Tree | - | 6 | - | - | - | - |
| | Burned | 6 | 6 | 6 | 6 | 6 | 6 |
| 2009 | Unburned | 6 | 6 | - | - | - | - |
| | Unburned-Downed Tree | - | 6 | - | - | - | - |
| | Burned | 6 | 6 | - | - | - | - |
| 2015 | Control | - | - | 5 | 5 | 5 | 5 |
| | Bullhog | - | - | - | - | 5 | 5 |
| | Burned | - | - | 5 | 5 | 5 | 5 |
| | Cut | - | - | 5 | 5 | 5 | 5 |
| | Cut-Downed Tree | - | - | - | 5 | - | 5 |

- Indicates not applicable, no plots.





**Table 3.** Select foliar cover and ground cover measures on from hillslope-scale site characterization plots (990 m²) in cut and burned treatment areas at the Marking Corral and Onaqui sites 1 yr prior to tree removal (2006) and 1 yr (2007) and 9 yr (2015) after tree removal treatments.

| | Marking Corral | | | Onaqui | | |
|---|---|---|---|---|---|---|
| Site characteristic | Untreated 2006[1] | Cut 2007[2] | Cut 2015[2] | Untreated 2006[1] | Cut 2007[2] | Cut 2015[2] |
| **Foliar Cover** | | | | | | |
| Shrub (%) | 14.6 | 14.3 | 28.7 | 3.4 | 5.0 | 16.9 |
| Grass (%) | 12.4 | 21.4 | 30.2 | 7.3 | 13.7 | 27.1 |
| Forb (%) | 1.0 | 3.7 | 1.4 | 3.2 | 12.1 | 7.4 |
| **Ground Cover** | | | | | | |
| Litter (%) | 46.1 | 46.0 | 47.6 | 26.2 | 41.6 | 35.8 |
| Rock (%)[3] | 22.0 | 11.3 | 1.3 | 29.8 | 22.3 | 17.0 |
| Bare soil (%) | 26.4 | 40.5 | 42.5 | 37.7 | 29.1 | 35.7 |

| | Marking Corral | | | Onaqui | | |
|---|---|---|---|---|---|---|
| Site characteristic | Untreated 2006[1] | Burn 2007[4] | Burn 2015[4] | Untreated 2006[1] | Burn 2007[4] | Burn 2015[4] |
| **Foliar Cover** | | | | | | |
| Shrub (%) | 17.7 | 6.2 | 8.7 | 0.9 | 0.4 | 10.7 |
| Grass (%) | 4.8 | 10.0 | 63.1 | 6.2 | 3.4 | 39.7 |
| Forb (%) | 0.1 | 10.6 | 0.9 | 3.3 | 6.0 | 14.3 |
| **Ground Cover** | | | | | | |
| Litter (%) | 47.4 | 31.4 | 40.3 | 34.4 | 29.7 | 34.7 |
| Rock (%)[3] | 25.4 | 16.5 | 12.8 | 29.0 | 31.6 | 21.6 |
| Bare soil (%) | 26.8 | 52.0 | 39.7 | 31.1 | 35.9 | 29.5 |

[1] Data from Pierson et al. (2010), but restricted to plots in area subsequently cut or burned at the respective site × treatment combination.
[2] Data from Williams et al. (2019a).
[3] Rock fragments > 5 mm in diameter.
[4] Data from Williams et al. (2018b).



**Table 4.** Soil texture and bulk density variables and data structure for those measures for all study sites.

| Site | Microsite | Percent Sand | Percent Silt | Percent Clay | Bulk Density (g cm$^{-3}$) |
|---|---|---|---|---|---|
| Castlehead | interspace | 50.4 | 43.7 | 5.9 | 1.04 |
| Castlehead | juniper_cop | 65.3 | 31.5 | 3.2 | 0.72 |
| Castlehead | shrub_cop | 61.8 | 34.6 | 3.6 | 0.76 |
| Marking Corral | interspace | 63.5 | 32.3 | 4.3 | 1.35 |
| Marking Corral | juniper_cop | 74.4 | 23.2 | 2.3 | 1.05 |
| Marking Corral | pinyon_cop | 68.4 | 28.3 | 3.4 | 1.1 |
| Marking Corral | shrub_cop | 59.9 | 35.4 | 4.7 | 1.14 |
| Onaqui | interspace | 57.4 | 36.2 | 6.5 | 1.07 |
| Onaqui | juniper_cop | 58.9 | 35.6 | 5.4 | 0.83 |
| Onaqui | shrub_cop | 56.2 | 36.9 | 6.9 | 1.02 |





**Table 5.** Example (subset) of vegetation and ground cover variables and data structure for measures on hillslope-scale site characterization plots (990 m$^2$) at the study sites.

| Plot_ID | Site | Year | Treatment Area | Treated Yes or No | Fol. Cvr. Shrub (%) | Fol. Cvr. Grass (%) | Fol. Cvr. Forb (%) | ... | Live Shrubs (>5 cm) Per Ha | Dead Shrub (>5 cm) per Ha | JUOC Trees (>0.5 m) Per Ha | JUOC Trees (5-50 cm) Per Ha |
|---|---|---|---|---|---|---|---|---|---|---|---|---|
| SC_CH_BURN1 | Castlehead | 2008 | Burn | Yes | 0 | 5.3 | 6.3 | ... | 0 | 722 | 0 | 0 |
| SC_CH_BURN2 | Castlehead | 2008 | Burn | Yes | 0 | 3.7 | 5.7 | ... | 0 | 611 | 0 | 0 |
| SC_CH_BURN3 | Castlehead | 2008 | Burn | Yes | 0 | 5 | 4 | ... | 0 | 1389 | 0 | 0 |
| SC_CH_UNB1 | Castlehead | 2008 | Unburned | No | 0 | 13.3 | 6.7 | ... | 222 | 278 | 222 | 5.5 |
| SC_CH_UNB2 | Castlehead | 2008 | Unburned | No | 4 | 26.3 | 6.7 | ... | 1944 | 778 | 162 | 4.7 |
| SC_CH_UNB3 | Castlehead | 2008 | Unburned | No | 14.7 | 12.3 | 6.3 | ... | 4056 | 1944 | 121 | 4.2 |
| SC_CH_BURN1 | Castlehead | 2009 | Burn | Yes | 0 | 22 | 17 | ... | 56 | 278 | 0 | 0 |
| SC_CH_BURN2 | Castlehead | 2009 | Burn | Yes | 0 | 12.7 | 25.3 | ... | 111 | 2500 | 0 | 0 |
| SC_CH_BURN3 | Castlehead | 2009 | Burn | Yes | 0 | 16.3 | 26.3 | ... | 0 | 1833 | 0 | 0 |
| SC_CH_UNB1 | Castlehead | 2009 | Unburned | No | 1 | 19.3 | 2 | ... | 5278 | 2056 | 212 | 5.9 |
| SC_CH_UNB2 | Castlehead | 2009 | Unburned | No | 14.7 | 46.3 | 7 | ... | 722 | 56 | 111 | 6.2 |
| SC_CH_UNB3 | Castlehead | 2009 | Unburned | No | 18.3 | 39 | 14.3 | ... | 5667 | 2056 | 121 | 4.6 |
| ... | ... | ... | ... | ... | ... | ... | ... | ... | ... | ... | ... | ... |
| SC_ON_CUT1 | Onaqui | 2015 | Cut | Yes | 8.9 | 41.6 | 11.3 | ... | 6389 | 0 | 0 | 0 |
| SC_ON_CUT2 | Onaqui | 2015 | Cut | Yes | 21 | 21 | 7.1 | ... | 10667 | 0 | 0 | 0 |
| SC_ON_CUT3 | Onaqui | 2015 | Cut | Yes | 20.8 | 18.7 | 3.9 | ... | 10611 | 0 | 0 | 0 |



**Table 6.** Example (subset) of rainfall simulation, vegetation, ground cover, and soil variables and data structure for measures on small-rainfall simulation plots (0.5 m²) at the study sites.

| Plot_ID | Site | Year | Treatment Area | Treated Yes or No | Microsite | Applied Rain DryRun (mm) | Applied Rain WetRun (mm) | Slope (%) | Random Roughness (mm) | ... | Fol. Cvr. Shrub (%) | Fol. Cvr. Grass (%) | Grd. Cvr. Bare Soil (%) | Grd. Cvr. Rock (%) | WDPT at 0 cm (s) | ... |
|---|---|---|---|---|---|---|---|---|---|---|---|---|---|---|---|---|
| SP_MC_CONT81 | Marking Corral | 2006 | Control | No | shrub_cop | 48 | 76 | 16.3 | 15 | ... | 51.4 | 25.7 | 23.8 | 8.9 | 3 | ... |
| SP_MC_CONT82 | Marking Corral | 2006 | Control | No | pinyon_cop | 48 | 76 | 22.2 | 26 | ... | 18.1 | 0 | 0 | 0 | 7 | ... |
| SP_MC_CONT83 | Marking Corral | 2006 | Control | No | interspace | 48 | 76 | 12.8 | 12 | ... | 0 | 54.3 | 20.7 | 47.8 | 3 | ... |
| SP_MC_CONT84 | Marking Corral | 2006 | Control | No | shrub_cop | 48 | 77 | 12.8 | 29 | ... | 78.1 | 19 | 13.6 | 4.9 | 3 | ... |
| SP_MC_CONT85 | Marking Corral | 2006 | Control | No | interspace | 48 | 76 | 20.7 | 12 | ... | 0 | 34.3 | 38.1 | 30.9 | 3 | ... |
| SP_MC_CONT86 | Marking Corral | 2006 | Control | No | pinyon_cop | 47 | 76 | 13.6 | 15 | ... | 0 | 2.9 | 0 | 0 | 3 | ... |
| SP_MC_CONT87 | Marking Corral | 2006 | Control | No | interspace | 44 | 77 | 9.6 | 14 | ... | 4.8 | 35.2 | 48.9 | 13 | 3 | ... |
| SP_MC_CONT88 | Marking Corral | 2006 | Control | No | shrub_cop | 45 | 76 | 11.2 | 22 | ... | 61.9 | 18.1 | 17 | 3.4 | 3 | ... |
| SP_MC_CONT89 | Marking Corral | 2006 | Control | No | juniper_cop | 48 | 78 | 9.9 | 10 | ... | 0 | 1.9 | 0 | 0 | 38 | ... |
| SP_MC_CONT90 | Marking Corral | 2006 | Control | No | juniper_cop | 48 | 78 | 17.5 | 12 | ... | 0 | 26.7 | 0 | 1 | 51 | ... |
| SP_MC_CUT91 | Marking Corral | 2006 | Cut | No | interspace | 48 | 78 | 9.5 | 4 | ... | 0 | 0 | 37.1 | 61.9 | 3 | ... |
| ... | ... | ... | ... | ... | ... | ... | ... | ... | ... | ... | ... | ... | ... | ... | ... | ... |
| SP_ON_CONT78 | Onaqui | 2015 | Control | No | shrub_cop | 46 | 74 | 18.1 | 17 | ... | 42.9 | 8.6 | 14.4 | 27.8 | 30 | ... |
| SP_ON_CONT79 | Onaqui | 2015 | Control | No | interspace | 46 | 74 | 19.2 | 13 | ... | 0 | 7.6 | 24.1 | 59.5 | 3 | ... |
| SP_ON_CONT80 | Onaqui | 2015 | Control | No | shrub_cop | 47 | 75 | 17.5 | 19 | ... | 72.4 | 12.4 | 31.6 | 16.8 | 3 | ... |





**Table 7.** Example (subset) of rainfall simulation, vegetation, ground cover, and soil variables and data structure for measures on large-rainfall simulation plots (13 m²) at the study sites.

| Plot_ID | Site | Year | Treatment Area | Treated Yes or No | Microsite | Applied Rain DryRun (mm) | Applied Rain WetRun (mm) | Slope (%) | Random Roughness (mm) | ... | Fol. Cvr. Shrub (%) | Fol. Cvr. Grass (%) | Grd. Cvr. Bare Soil (%) | Grd. Cvr. Rock (%) | Avg. Canopy Gap (cm) | Avg. Basal Gap (cm) |
|---|---|---|---|---|---|---|---|---|---|---|---|---|---|---|---|---|
| LP_MC_CUT37 | Marking Corral | 2006 | Cut | No | juniper_cop | 39 | 65 | 11.1 | 19 | ... | 9.2 | 8.8 | 16.3 | 6.1 | 100 | 164 |
| LP_MC_CUT38 | Marking Corral | 2006 | Cut | No | juniper_cop | 47 | 87 | 12.2 | 19 | ... | 7.5 | 8.8 | 5.4 | 2 | 77 | 157 |
| LP_MC_CUT39 | Marking Corral | 2006 | Cut | No | intercanopy | 37 | 63 | 10.4 | 18 | ... | 29.8 | 18 | 44.1 | 5.4 | 83 | 121 |
| LP_MC_CUT40 | Marking Corral | 2006 | Cut | No | intercanopy | 50 | 96 | 9.6 | 14 | ... | 11.5 | 8.8 | 28.1 | 49.2 | 59 | 94 |
| LP_MC_CUT41 | Marking Corral | 2006 | Cut | No | intercanopy | 40 | 67 | 8.8 | 18 | ... | 21.4 | 13.9 | 27.2 | 22.8 | 76 | 125 |
| LP_MC_CUT42 | Marking Corral | 2006 | Cut | No | intercanopy | 46 | 88 | 9.5 | 15 | ... | 18.6 | 19.7 | 24.1 | 31.2 | 86 | 131 |
| LP_MC_CUT43 | Marking Corral | 2006 | Cut | No | pinyon_cop | 39 | 72 | 9.3 | 20 | ... | 0.3 | 3.4 | 0 | 0.7 | 428 | 499 |
| LP_MC_CUT44 | Marking Corral | 2006 | Cut | No | pinyon_cop | 50 | 93 | 8.1 | 18 | ... | 0.7 | 1.7 | 0 | 1 | 427 | 435 |
| LP_MC_CUT45 | Marking Corral | 2006 | Cut | No | pinyon_cop | 46 | 94 | 9.1 | 15 | ... | 10.8 | 4.4 | 0 | 1.4 | 113 | 168 |
| LP_MC_CUT46 | Marking Corral | 2006 | Cut | No | pinyon_cop | 47 | 83 | 13 | 21 | ... | 9.8 | 8.5 | 1 | 3.7 | 127 | 243 |
| LP_MC_CUT47 | Marking Corral | 2006 | Cut | No | intercanopy | 41 | 80 | 12.1 | 25 | ... | 26.9 | 19.4 | 32.9 | 29.5 | 69 | 110 |
| ... | ... | ... | ... | ... | ... | ... | ... | ... | ... | ... | ... | ... | ... | ... | ... | ... |
| LP_CH_BURN28 | Castlehead | 2008 | Burn | Yes | juniper_cop | 43 | 77 | 15.2 | 12 | ... | 0 | 3.7 | 33.6 | 50.2 | 43 | 101 |
| LP_CH_BURN29 | Castlehead | 2008 | Burn | Yes | intercanopy | 48 | 87 | 14.8 | 22 | ... | 0 | 5.8 | 54.2 | 44.7 | 36 | 56 |
| LP_CH_BURN30 | Castlehead | 2008 | Burn | Yes | intercanopy | 42 | 83 | 14.7 | 17 | ... | 0 | 6.8 | 33.6 | 54.2 | 31 | 42 |





**Table 8.** Example (subset) of overland flow, vegetation, and ground cover variables and data structure for measures on overland flow simulation plots (~9 m²) at the study sites.

| Plot_ID | Site | Year | Treatment Area | Treated Yes or No | Microsite | Avg. Width 15 L min⁻¹ at 3m (cm) | Avg. Width 30 L min⁻¹ at 3m (cm) | Avg. Width 45 L min⁻¹ at 3m (cm) | ... | Avg. Velocity 15 L min⁻¹ (m s⁻¹) | Avg. Velocity 30 L min⁻¹ (m s⁻¹) | Avg. Velocity 45 L min⁻¹ (m s⁻¹) | Avg. Canopy Gap (cm) | Avg. Basal Gap (cm) |
|---|---|---|---|---|---|---|---|---|---|---|---|---|---|---|
| RI_MC_CUT37 | Marking Corral | 2006 | Cut | No | juniper_cop | 2 | 10 | 28 | ... | -999 | 0.029 | 0.036 | 67 | 92 |
| RI_MC_CUT38 | Marking Corral | 2006 | Cut | No | juniper_cop | 0 | 30 | 32 | ... | 0 | -999 | 0.058 | 78 | 156 |
| RI_MC_CUT39 | Marking Corral | 2006 | Cut | No | intercanopy | 42 | 33 | 43 | ... | 0.07 | 0.122 | 0.148 | 70 | 93 |
| RI_MC_CUT40 | Marking Corral | 2006 | Cut | No | intercanopy | 50 | 38 | 53 | ... | 0.085 | 0.127 | 0.131 | 55 | 100 |
| RI_MC_CUT41 | Marking Corral | 2006 | Cut | No | intercanopy | 37 | 61 | 59 | ... | 0.028 | 0.067 | 0.107 | 59 | 106 |
| RI_MC_CUT42 | Marking Corral | 2006 | Cut | No | intercanopy | 47 | 61 | 52 | ... | 0.05 | 0.066 | 0.1 | 86 | 109 |
| RI_MC_CUT43 | Marking Corral | 2006 | Cut | No | pinyon_cop | 0 | 52 | 102 | ... | 0 | -999 | 0.038 | 333 | 333 |
| RI_MC_CUT44 | Marking Corral | 2006 | Cut | No | pinyon_cop | 0 | -999 | -999 | ... | 0 | -999 | -999 | 284 | 292 |
| RI_MC_CUT45 | Marking Corral | 2006 | Cut | No | pinyon_cop | 0 | 0 | 0 | ... | 0 | 0 | -999 | 131 | 172 |
| RI_MC_CUT46 | Marking Corral | 2006 | Cut | No | pinyon_cop | 0 | 24 | 32 | ... | 0 | 0.033 | 0.044 | 88 | 175 |
| RI_MC_CUT47 | Marking Corral | 2006 | Cut | No | intercanopy | 64 | 64 | 52 | ... | 0.062 | 0.098 | 0.127 | 79 | 85 |
| ... | ... | ... | ... | ... | ... | ... | ... | ... | ... | ... | ... | ... | ... | ... |
| RI_ON_CUT131 | Onaqui | 2015 | Cut | Yes | intercanopy | 144 | 148 | 158 | ... | 0.051 | 0.084 | 0.182 | 46 | 46 |
| RI_ON_CUT133 | Onaqui | 2015 | Cut | Yes | intercanopy | 0 | 165 | 82 | ... | 0 | 0.054 | 0.073 | 65 | 34 |
| RI_ON_CUT134 | Onaqui | 2015 | Cut | Yes | intercanopy | 0 | 29 | 36 | ... | 0 | 0.062 | 0.086 | 48 | 58 |



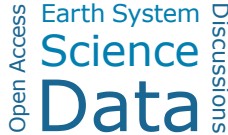
**Table 9.** Example (subset) of time series runoff and sediment data from small-plot rainfall simulations (0.5 m$^2$) at the study sites.

| Plot_ID | Site | Year | Treatment Area | Treated Yes or No | Microsite | Run Type | Runoff Yes or No | Rainfall Rate (mm h$^{-1}$) | Runoff Start Time (mm:ss) | Simulation Time (mm:ss) | Sample Fill Time (s) | Runoff (L min$^{-1}$) | Sediment Conc. (g L$^{-1}$) | Runoff (mm h$^{-1}$) | Sediment Discharge (g s$^{-1}$) |
|---|---|---|---|---|---|---|---|---|---|---|---|---|---|---|---|
| SP_MC_CONT81 | Marking Corral | 2006 | Control | No | shrub_cop | Dry_Run | No | 64 | | 00:00 | 0 | 0.000 | 0.00 | 0.000 | 0.0000 |
| SP_MC_CONT81 | Marking Corral | 2006 | Control | No | shrub_cop | Dry_Run | No | 64 | | 44:00 | 0 | 0.000 | 0.00 | 0.000 | 0.0000 |
| SP_MC_CONT81 | Marking Corral | 2006 | Control | No | shrub_cop | Wet_Run | Yes | 102 | 05:11 | 00:00 | 0 | 0.000 | 0.00 | 0.000 | 0.0000 |
| SP_MC_CONT81 | Marking Corral | 2006 | Control | No | shrub_cop | Wet_Run | Yes | 102 | 05:11 | 05:10 | 0 | 0.000 | 0.00 | 0.000 | 0.0000 |
| SP_MC_CONT81 | Marking Corral | 2006 | Control | No | shrub_cop | Wet_Run | Yes | 102 | 05:11 | 05:36 | 49 | 0.096 | 0.38 | 11.520 | 0.0006 |
| SP_MC_CONT81 | Marking Corral | 2006 | Control | No | shrub_cop | Wet_Run | Yes | 102 | 05:11 | 06:30 | 60 | 0.095 | 0.10 | 11.436 | 0.0002 |
| SP_MC_CONT81 | Marking Corral | 2006 | Control | No | shrub_cop | Wet_Run | Yes | 102 | 05:11 | 07:30 | 60 | 0.080 | 0.13 | 9.552 | 0.0002 |
| SP_MC_CONT81 | Marking Corral | 2006 | Control | No | shrub_cop | Wet_Run | Yes | 102 | 05:11 | 08:30 | 60 | 0.074 | 0.00 | 8.840 | 0.0000 |
| SP_MC_CONT81 | Marking Corral | 2006 | Control | No | shrub_cop | Wet_Run | Yes | 102 | 05:11 | 09:30 | 60 | 0.068 | 0.30 | 8.110 | 0.0003 |
| ... | ... | ... | ... | ... | ... | ... | ... | ... | ... | ... | ... | ... | ... | ... | ... |
| SP_ON_CONT80 | Onaqui | 2015 | Control | No | shrub_cop | Wet_Run | Yes | 100 | 01:50 | 26:00 | 120 | 0.075 | 6.42 | 8.968 | 0.0080 |
| SP_ON_CONT80 | Onaqui | 2015 | Control | No | shrub_cop | Wet_Run | Yes | 100 | 01:50 | 29:00 | 120 | 0.077 | 6.29 | 9.250 | 0.0081 |
| SP_ON_CONT80 | Onaqui | 2015 | Control | No | shrub_cop | Wet_Run | Yes | 100 | 01:50 | 32:00 | 120 | 0.073 | 6.58 | 8.751 | 0.0080 |
| SP_ON_CONT80 | Onaqui | 2015 | Control | No | shrub_cop | Wet_Run | Yes | 100 | 01:50 | 35:00 | 120 | 0.065 | 6.55 | 7.783 | 0.0071 |
| SP_ON_CONT80 | Onaqui | 2015 | Control | No | shrub_cop | Wet_Run | Yes | 100 | 01:50 | 41:00 | 120 | 0.067 | 6.50 | 8.026 | 0.0073 |





**Table 10.** Example (subset) of time series runoff and sediment data from large-plot rainfall simulations (13 m$^2$) at the study sites.

| Plot_ID | Site | Year | Treatment Area | Treated Yes or No | Microsite | Downed Cut Tree Yes or No | Run Type | Rainfall Rate (mm h⁻¹) | Runoff Start Time (mm:ss) | Simulation Time (mm:ss) | Sample Fill Time (s) | Runoff (L min⁻¹) | Sediment Conc. (g L⁻¹) | Runoff (mm h⁻¹) | Sediment Discharge (g s⁻¹) |
|---|---|---|---|---|---|---|---|---|---|---|---|---|---|---|---|
| LP_MC_CUT37 | Marking Corral | 2006 | Cut | No | juniper_cop | No | Dry_Run | 52 | 08:15 | 00:00 | 0 | 0 | 0 | 0 | 0 |
| LP_MC_CUT37 | Marking Corral | 2006 | Cut | No | juniper_cop | No | Dry_Run | 52 | 08:15 | 08:14 | 0 | 0 | 0 | 0 | 0 |
| LP_MC_CUT37 | Marking Corral | 2006 | Cut | No | juniper_cop | No | Dry_Run | 52 | 08:15 | 09:05 | 20 | 0.294 | 19.08 | 1.357 | 0.094 |
| LP_MC_CUT37 | Marking Corral | 2006 | Cut | No | juniper_cop | No | Dry_Run | 52 | 08:15 | 10:08 | 15 | 0.464 | 14.56 | 2.142 | 0.113 |
| LP_MC_CUT37 | Marking Corral | 2006 | Cut | No | juniper_cop | No | Dry_Run | 52 | 08:15 | 12:08 | 15 | 0.627 | 8.74 | 2.894 | 0.091 |
| LP_MC_CUT37 | Marking Corral | 2006 | Cut | No | juniper_cop | No | Dry_Run | 52 | 08:15 | 14:08 | 16 | 0.476 | 11.11 | 2.196 | 0.088 |
| LP_MC_CUT37 | Marking Corral | 2006 | Cut | No | juniper_cop | No | Dry_Run | 52 | 08:15 | 16:08 | 15 | 0.625 | 10.69 | 2.883 | 0.111 |
| LP_MC_CUT37 | Marking Corral | 2006 | Cut | No | juniper_cop | No | Dry_Run | 52 | 08:15 | 18:08 | 15 | 0.554 | 10.47 | 2.556 | 0.097 |
| LP_MC_CUT37 | Marking Corral | 2006 | Cut | No | juniper_cop | No | Dry_Run | 52 | 08:15 | 20:08 | 15 | 0.609 | 12.21 | 2.812 | 0.124 |
| ⋮ | ⋮ | ⋮ | ⋮ | ⋮ | ⋮ | ⋮ | ⋮ | ⋮ | ⋮ | ⋮ | ⋮ | ⋮ | ⋮ | ⋮ | ⋮ |
| LP_CH_BURN30 | Castlehead | 2008 | Burn | Yes | intercanopy | No | Wet_Run | 110 | 01:09 | 30:08 | 15 | 15.647 | 4.68 | 72.216 | 1.22 |
| LP_CH_BURN30 | Castlehead | 2008 | Burn | Yes | intercanopy | No | Wet_Run | 110 | 01:09 | 33:08 | 15 | 13.819 | 4.41 | 63.781 | 1.015 |
| LP_CH_BURN30 | Castlehead | 2008 | Burn | Yes | intercanopy | No | Wet_Run | 110 | 01:09 | 36:08 | 15 | 14.198 | 5.78 | 65.529 | 1.368 |
| LP_CH_BURN30 | Castlehead | 2008 | Burn | Yes | intercanopy | No | Wet_Run | 110 | 01:09 | 39:08 | 15 | 16.666 | 5.65 | 76.919 | 1.569 |
| LP_CH_BURN30 | Castlehead | 2008 | Burn | Yes | intercanopy | No | Wet_Run | 110 | 01:09 | 42:08 | 15 | 14.282 | 5.48 | 65.915 | 1.305 |





**Table 11.** Example (subset) of time series runoff and sediment data from overland flow simulations (~9 m²) at the study sites.

| Plot_ID | Site | Year | Treatment Area | Treated Yes or No | Microsite | Plot Bordered All Sides Yes or No | Runoff 15 L min⁻¹ Yes or No | Runoff 30 L min⁻¹ Yes or No | Runoff 45 L min⁻¹ Yes or No | Applied Overland Flow Rate (L min⁻¹) | Simulation Time (mm:ss) | Sample Fill Time (s) | Runoff (L min⁻¹) | Sediment Conc. (g L⁻¹) |
|---|---|---|---|---|---|---|---|---|---|---|---|---|---|---|
| R1_MC_CUT37 | Marking Corral | 2006 | Cut | No | juniper_cop | Yes | Yes | Yes | Yes | 15 | 00:00 | 30 | 0.181 | 13.49 |
| R1_MC_CUT37 | Marking Corral | 2006 | Cut | No | juniper_cop | Yes | Yes | Yes | Yes | 15 | 00:41 | 15 | 0.47 | 1.62 |
| R1_MC_CUT37 | Marking Corral | 2006 | Cut | No | juniper_cop | Yes | Yes | Yes | Yes | 15 | 01:11 | 15 | 0.628 | 0.7 |
| R1_MC_CUT37 | Marking Corral | 2006 | Cut | No | juniper_cop | Yes | Yes | Yes | Yes | 15 | 02:31 | 15 | 1.265 | 0.66 |
| R1_MC_CUT37 | Marking Corral | 2006 | Cut | No | juniper_cop | Yes | Yes | Yes | Yes | 15 | 03:06 | 15 | 1.662 | 1.04 |
| R1_MC_CUT37 | Marking Corral | 2006 | Cut | No | juniper_cop | Yes | Yes | Yes | Yes | 15 | 03:41 | 15 | 1.976 | 0.2 |
| R1_MC_CUT37 | Marking Corral | 2006 | Cut | No | juniper_cop | Yes | Yes | Yes | Yes | 30 | 00:00 | 15 | 11.181 | 15.97 |
| R1_MC_CUT37 | Marking Corral | 2006 | Cut | No | juniper_cop | Yes | Yes | Yes | Yes | 30 | 00:45 | 15 | 14.551 | 0.61 |
| R1_MC_CUT37 | Marking Corral | 2006 | Cut | No | juniper_cop | Yes | Yes | Yes | Yes | 30 | 02:40 | 15 | 18.795 | 0.29 |
| ⋮ | ⋮ | ⋮ | ⋮ | ⋮ | ⋮ | ⋮ | ⋮ | ⋮ | ⋮ | ⋮ | ⋮ | ⋮ | ⋮ | ⋮ |
| R1_ON_CUT134 | Onaqui | 2015 | Cut | Yes | intercanopy | No | No | Yes | Yes | 45 | 04:05 | 20 | 14.5 | 5.51 |
| R1_ON_CUT134 | Onaqui | 2015 | Cut | Yes | intercanopy | No | No | Yes | Yes | 45 | 04:55 | 20 | 15.215 | 5.56 |
| R1_ON_CUT134 | Onaqui | 2015 | Cut | Yes | intercanopy | No | No | Yes | Yes | 45 | 05:45 | 20 | 15.694 | 5.49 |
| R1_ON_CUT134 | Onaqui | 2015 | Cut | Yes | intercanopy | No | No | Yes | Yes | 45 | 08:35 | 20 | 17.426 | 5.41 |
| R1_ON_CUT134 | Onaqui | 2015 | Cut | Yes | intercanopy | No | No | Yes | Yes | 45 | 10:35 | 20 | 18.678 | 5.44 |