# Peer review of "Vegetation, ground cover, soil, rainfall simulation, and overland flow experiments before and after tree removal in woodlandencroached sagebrush steppe: the hydrology component of the Sagebrush Steppe Treatment Evaluation Project (SageSTEP)"

_Earth System Science Data, 2019_

## Referee Comment (RC1) · Anonymous Referee #1 · 28 Jan 2020

Williams et al. provide an important and highly valuable contribution to plot-scale experiments on runoff and soil erosion in the semi-arid Great Basin, USA. Up to my best knowledge, this is the most extensive data set currently available. These data are important to parameterize commonly applied runoff and soil erosion models, such as RHEM. While I am convinced that this data set is highly relevant for a wide array of

scientific disciplines for model applications and hypothesis testing, there are several suggestions, I'd like to point out: - The main section on 'Study sites and Experimental design' is hard to follow. Maybe the authors could better link the descriptions to Table 2 provided in the manuscript.

- In the field methods section, the authors did explain how foliage is estimated. I was wondering if the foliage is as static as described here or if foliage does differ over the seasons? In that case, additional information on the season the experiment was conducted should be provided.

- The applied rainfall intensities are assumed to reflect the natural rainfall distributions. However, the data from rain gauges close to the experimental sites is not shown. I suggest to include such a graph. It is well established that rainfall simulations often exceed natural rainfall intensities, sometimes up to an order of magnitude. This conflict complicates the transfer from small-scale findings to natural systems, e.g. modeling studies often on a larger spatial scale. Regardless, the authors should better explain their choice of rainfall intensities. Sometimes higher-than-natural intensities are intentionally chosen to amplify hydrological responses on diverse environmental settings.

- The authors state that 'wet' simulations are conducted on plots where rainfall was applied for the previous dry runs. The time lag between both runs (dry vs. wet) is 30 min (lines 274-275). While I see the general and often unavoidable restrictions with such difficult and comprehensive experiments, I was wondering if this experimental design is really appropriate. Given the first dry run preceding the wet run, one could expect that all fine, and thus, mobile soil sediment has been evacuated during the dry run and, consequently, the wet runs may be more supply limited than the previous dry run. Did the authors account for such potential shift in the soil erosion regime? The authors could, for example, provide exemplary sediment hysteresis to test for this. I am convinced that such a graph would add a lot of relevant information.

- By inspecting the data sets available for downloading, I saw that many of the exper-
iments were restricted to 45 minutes (e.g. small_time_series-csv). May the authors explain such time restriction?

- Lastly, while I highly appreciate the efforts the authors put into the generation of this data set, I was wondering how these data relate to previous studies conducted in other study areas but the ones presented here. Do the authors see the chance to use and/or transfer their data set for studies outside the Great Basin area?

---

## Referee Comment (RC2) · Anonymous Referee #2 · 23 Feb 2020

The manuscript presents extensive data on numerous parameters characterizing surface and shallow subsurface hydrology at three locations within the western U.S. These data are concise and relevant for future hydrological and sedimentary analysis, and potential inclusion to various land surface models. The manuscript is available for download via the URL provided by the authors.

1. The description of plot scales should be consistent throughout the manuscript. In the Abstract, only 'overland flow' plots are mentioned explicitly; this changes to rainfall simulations at various plot sizes and overland flow plots in Lines 111-113, and finally to four plot scales in Lines 148-150, hillslope plots added. Besides, a small figure showing locations for each plot could be useful for non-U.S. readership.

2. This inconsistency is brought further to the text, Section 3, where field methods description starts with hillslope-scale plots, the largest, and continues with small- and large-scale plots etc. Though there might be a certain logic in such description order, I would suggest to follow either top-down or bottom-up approach.

3. Lines 287-288, the sediment concentration is said to be calculated from runoff samples by weighing; what is a 'runoff sample'? Is it a liquid volume - and if yes, was it just dried to full sample evaporation? If not, was any filtration system used, and if yes, then what were its parameters - pore size etc?

3. The dataset is well-organized, but several technical corrections are needed: 3.1. Data Dictionary - data types should be presented as standard notation, i.e. integer, real, character etc; same, variable sizes should be given, i.e. as INT/LONG INT/DOUBLE/CHAR(X) etc. 3.2. Categorical variables are multiple in the Data Dictionary, and are particularly poorly described; possible categories are listed as 'Acceptable values', which is not the best way to present them. No explanation on whet does, e.g. 'Tracked_LowMulch' mean, is given in the dataset itself. A separate table explaining your categorical variables is needed, or you might suggest a better way of presentation. 3.3. Same, 'Yes/No' is not a character variable, but has LOGICAL type, therefore acceptable values are 0/1, Y/N, or T/F, each is valid. 3.4. Dataset contains some info on treatment area and date, but I've found no clear descriptors for treatment type for each dataset in the plot characteristics table. This raises the question on whether the variables are correctly distributed between various dataset tables. 3.5. Table 3 contains no info on either plot type (small vs large vs overland etc) or plot area. 3.6. I find it difficult to browse through data with visual inspection, since: PLOT_ID is a

last column, e.g. in Table 4, and is hard to find in other tables as well; in several tables, PLOT_ID is not unique since two rows contain data for differerent years; treatment date repeats in Tables 3 and 4. In general, column sequence is not entirely logical, and can be enhanced.

The dataset structure, I believe, should be subject to technical inspection. I suggest the authors to read your dataset to R/RStudio environment and check dataset usability / statistical analysis performance.

---

## Referee Comment (RC3) · Anonymous Referee #3 · 26 Feb 2020

General comments:

Authors present extensive and detailed dataset with vegetation, ground cover, soils, hydrology, and erosion data from over 1000 plots in diverse vegetation, ground cover, and surface soil conditions from three study sites in USA for five study years. Presented data is of high scientific importance and probable usage in the future. Study

sites, experimental design and field methods are well described. There are no explicit estimates of the data errors and its discussion. Consider adding some uncertainty estimates in the Section 2 or Section 3.

Paper does not provide information about which exactly kind of data is in the dataset. Reader is not able to decide whether he/she interested to download data or not based just on the paper. I suggest including a new section or subsection or extend Section 5 and include brief technical overview of the data covering description of variables from the dataset (maybe in a table that is shorter version of the table "SageSTEP_Database_Data_Dictionary" from the dataset), technical details (could be from lines 450-461) and structure of the data files.

Section 4 is important for understanding of scientific significance of the presented dataset but lacks any scientific conclusions. It explains the previous usage of data. It would be good for readers to know not only descriptions of data usage but also the scientific results. I suggest expanding the section, brief presenting significant findings of the mentioned studies and referring to the Figures 3-5.

Specific comments:

Table 1: Intercanopy bare ground includes shrubs and grasses?

Table 2: There are 4 parts of the Table. What do they refer to? Consider adding informative titles to different parts of the table and relocate extensive description of different types of sites to the paper text.

Line 206: Are site characterization plots representative for all plots at each of three study sites?

Lines 450-459: Consider to relocate this detailed description of the dataset from the Conclusions to Section 5 or new Section / subsection with the technical overview of the data.

Data table "Small time series": Please explain what empty cells mean, for example

lines No 6099, 7431, 7504, 8349 of the columns "Runoff_L_min", "SedConc_g_L", "Runoff_mm_hr" and "SedDisch_g_s".

Technical corrections:

Link to the data DOI in the abstract and Section 5 leads to DOI Not Found webpage.

Line 387-389: It would be useful to show TRAW and width variables on the photo or on the scheme.

Figure 3: Do (a) and (c) refer to Marking Corral site and (b) and (d) – to Onaqui site? It should be explicitly noted in the Figure caption. Untreated tree coppice microsite indicated as bold green line in the legend but dash line on the graph. It would be better to use bold lines for all three control microsites.

Table 5-6: expand abbreviations Fol. Cvr., JUOC and WDPT.

---

## Author Comment (AC1) · 28 Mar 2020

ESSD-2019-182 - AUTHORS' RESPONSES TO REVIEWER COMMENTS ON " VEGETATION, GROUND COVER, SOIL, RAINFALL SIMULATION, AND OVERLAND FLOW EXPERIMENTS BEFORE AND AFTER TREE REMOVAL IN WOODLAND-

[Figure]

ENCROACHED SAGEBRUSH STEPPE: THE HYDROLOGY COMPONENT OF THE SAGEBRUSH STEPPE TREATMENT EVALUATION PROJECT (SAGESTEP)"

- Page and line number references in author responses are to the revised manuscript unless otherwise noted.

-Author specific responses to Reviewer comments are numbered sequentially relative to the entire list of responses.

RESPONSES TO ANONYMOUS REFEREE #1 COMMENTS:

1. Williams et al. provide an important and highly valuable contribution to plot-scale experiments on runoff and soil erosion in the semi-arid Great Basin, USA. Up to my best knowledge, this is the most extensive data set currently available. These data are important to parameterize commonly applied runoff and soil erosion models, such as RHEM. While I am convinced that this data set is highly relevant for a wide array of scientific disciplines for model applications and hypothesis testing, there are several suggestions [that] I'd like to point out.

Author Response: The authors appreciate the comments here regarding the importance and value of this extensive dataset and its potential applications. We also thank the reviewer for the numerous suggestions, addressed below in respective responses.

2. The main section on 'Study sites and Experimental design' is hard to follow. Maybe the authors could better link the descriptions to Table 2 provided in the manuscript.

Authors' Response: Label omissions in Table 2 (for each of the black filled rows) of the original submission are the source of the confusion noted here by Reviewer 1. Reviewers 2 and 3 also pointed out confusion with Table 2 linkage to the "Study Sites and Experimental Design" section. Reviewer 3 specifically noted the lack of labels for the four sections in Table 2 (see comment #24 below). We provide a corrected Table 2 in revision, with the labels for each section. The same table is archived in the correct form (with all labels) with the original dataset at the required data repository (https://doi.org/10.15482/USDA.ADC/1504518), US Department of Agriculture, National Agricultural Library, Ag Data Commons website (https://data.nal.usda.gov/).

Each of the four sections in Table 2 (separated by the black filled rows) is associated with a specific plot type (site characterization vegetation plots, small rainfall plots, large rainfall plots, or overland flow plots) and provides the number of respective plots by plot type for each study site for each Year × Treatment × Microsite combination. Table 2 therefore provides an overview of the study design across sites for each of the plot types and is important in understanding the information presented in the "Study Sites and Experimental Design" section. Collectively, Figures 1 and 2 provide visual examples of the rainfall simulation experiments, instrumentation, and associated microsites, and Tables 1 and 2, respectively, provide site descriptions for the three sites and the distribution of plots by plot type across study years and treatments. The omissions of labels in the black rows (for the sections) of Table 2 greatly affect this linkage. The corrected Table 2 provides the necessary clarity and linkage requested here by Reviewer 1 and the other reviewers in subsequent comments below (comment #9 by Reviewer 2 and comment #24 by Reviewer 3).

3. In the field methods section, the authors did explain how foliage is estimated. I was wondering if the foliage is as static as described here or if foliage does differ over the seasons? In that case, additional information on the season the experiment was conducted should be provided.

Authors' Response: All measurements were made in the summer season each sampling year, but that was not explicitly stated in the original manuscript. In revision, we replaced the text "The data were collected in years 2006-2015. . ." with "All data were collected in summer months in the years 2006-2015. . ." to clarify the season of measurement, as suggested by Reviewer 1 here. This revision is located at Lines 131-132 in the revised manuscript.

Foliage can vary across seasons on rangelands and at the sites in this study, as suggested by Reviewer 1 here. Our experiments provide foliage measures taken at the same point in time (summer season) as the hydrology and erosion experiments to address the controls/drivers of hydrologic and erosion responses and to assess the impacts of tree removal on vegetation as measured in the summer season. The research was not meant to characterize seasonal variation in foliage throughout each of the study years.

4. The applied rainfall intensities are assumed to reflect the natural rainfall distributions. However, the data from rain gauges close to the experimental sites is not shown. I suggest to include such a graph. It is well established that rainfall simulations often exceed natural rainfall intensities, sometimes up to an order of magnitude. This conflict complicates the transfer from small-scale findings to natural systems, e.g. modeling studies often on a larger spatial scale. Regardless, the authors should better explain their choice of rainfall intensities. Sometimes higher-than-natural intensities are intentionally chosen to amplify hydrological responses on diverse environmental settings.

Authors' Response: As noted by Reviewer 1, rainfall intensities in rainfall simulation experiments are typically applied at rates intended to exceed infiltration capacity and generate runoff. Without runoff, the infiltration capacity before runoff generation remains unknown, and predictive utility of the data is somewhat limited. Further, treatment effects studies, such as this one, commonly select rainfall rates that stress the system of study in order to evaluate treatment effectiveness in buffering runoff and erosion. Our selection of rainfall rates was based on these typical experimental requirements of rainfall simulation studies, which are well documented in the literature (as noted by the reviewer; see response to comment #5 below for list of studies with similar methodologies). We provided return intervals for rainfall events in our previous papers on the dataset, but omitted them here in attempt to limit duplication of text from our previous papers describing the methods. The journal editors required us to minimize repeating methodologies explicitly described in our associated publications on the experiments, and to, instead, simply cite those studies. We therefore provided references to the

original papers that contain these details (see Lines 272-277 in the original and revised manuscripts). The rainfall intensity for the dry-run simulations over 5-min, 10-min, and 15-min durations is equivalent to respective storm return intervals of 7 yr, 15 yr, and 25 yr. The wet –run intensity over 5-min, 10-min, and 15-min durations is equivalent to respective storm return intervals of 25 yr, 60 yr, and 120 yr. These return intervals are based on the NOAA precipitation-frequency atlas of the United States (NOAA Atlas 14, Volume 1, Version 4.0) (as cited in Pierson et al., 2010 and other publications noted at Lines 272-277 in the original manuscript). There are no rainfall gauges at the study sites specifically for intensity derivations. Of most importance for users of the data in modelling is knowledge of the rainfall rate applied; the plot vegetation, ground cover, and soil characteristics; and runoff/erosion rates. All of these measures are provided in the various tables. We are willing to add the above rainfall return-interval information to the revised manuscript if desired by the journal editors pending their requirements to limit duplication of methods specificity from the long list of associated publications on the experiments (see also response to comment #22 below for list of publications from the dataset).

5. The authors state that 'wet' simulations are conducted on plots where rainfall was applied for the previous dry runs. The time lag between both runs (dry vs. wet) is 30 min (lines 274-275). While I see the general and often unavoidable restrictions with such difficult and comprehensive experiments, I was wondering if this experimental design is really appropriate. Given the first dry run preceding the wet run, one could expect that all fine, and thus, mobile soil sediment has been evacuated during the dry run and, consequently, the wet runs may be more supply limited than the previous dry run. Did the authors account for such potential shift in the soil erosion regime? The authors could, for example, provide exemplary sediment hysteresis to test for this. I am convinced that such a graph would add a lot of relevant information.

Authors' Response: The dry- and wet-run methodologies applied in our experiments are common for rainfall simulation studies, largely for logistical reasons as pointed out

by Reviewer 1. The multiple intensities on a plot allow for more replications and for assessment of responses across different rainfall rates and/or soil wetness conditions without additional laborious installations, plot characterizations, and moving/setting up of rainfall simulators. The list of studies utilizing such efficiencies is long (abbreviated list spanning five decades is provided here, excludes papers by authors associated with this study: Blackburn, 1975; Roundy et al., 1978; Johnson and Blackburn, 1989; Simanton et al., 1991; Johansen et al., 2001; Pierson et al., 2002; Stone et al., 2008; Polyakov et al., 2018).

As Reviewer 1 points out, some wet-run erosion rates in this study may have been affected by respective dry-run simulations with runoff (typical to these type of experiments). In some cases, dry-run simulations yielded zero runoff and likely posed little to no impact on wet-run sediment discharge rates (with exception of wetter soils). Rainfall simulation data reported in tables for our experiments do not account for carryover effects between dry-run and wet-run simulations. Our data tables do report whether runoff occurred for the dry- and wet-run simulations for each rainfall simulation plot. Dry-run simulation carryover effects on erosion from wet-runs could be modelled on that basis, as suggested by Reviewer 1. However, we simply elect to provide full description of the methodologies employed and the actual time series data. Our goal is to provide an extensive dataset and allow users to utilize the dataset for respective applications, rather than to provide a full suite of analyses of the data. Our approach here is consistent with a similar extensive rainfall simulation database recently published in ESSD by Polyakov et al. (2018). It is impractical to conceive of all possible uses and applications and to account for all respective potential data amendments. Given the methodological explanations, a user can opt to include or exclude various components of the dataset as appropriate for the associated application, inclusive of any analyses required for such an assessment.

We have added the following text, at Lines 440-445 in the revised manuscript, to ensure data users recognize the potential for carryover effects from the dry-run to wet-run

simulations and for overland flow experiments:

"Time series runoff and sediment data provided for rainfall simulations and overland flow experiments do not account for carryover effects from one plot run to the next on a given plot in a given year (i.e., dry-run effects on wet-run simulations; effects of 15 L min-1 overland flow releases on subsequent 30-45 L min-1 overland flow releases). Data users should consider whether carryover effects impact respective applications and make applicable adjustments to acquired data."

Blackburn, W. H. (1975), Factors influencing infiltration and sediment production of semiarid rangelands in Nevada. Water Resources Research, 11(6), 929-937.

Johansen, M. P., Hakonson, T. E., & Breshears, D. D. (2001), Post-fire runoff and erosion from rainfall simulation: Contrasting forests with shrublands and grasslands. Hydrological Processes, 15(15), 2953-2965.

Johnson, C. W., & Blackburn, W. H. (1989), Factors contributing to sagebrush rangeland soil loss. Transactions of the American Society of Agricultural Engineers, 32(1), 155-160.

Pierson, F. B., Spaeth, K. E., Weltz, M. A., & Carlson, D. H. (2002), Hydrologic response of diverse western rangelands. Journal of Range Management, 55(6), 558-570.

Polyakov, V., Stone, J., Collins, C. H., Nearing, M. A., Paige, G., Buono, J., & Gomez-Pond, R. L. (2018), Rainfall simulation experiments in the southwestern USA using the Walnut Gulch Rainfall Simulator. Earth System Science Data, 10(1), 19-26. doi: 10.5194/essd-10-19-2018.

Roundy, B. A., Blackburn, W. H., & Eckert R.E., J. (1978), Influence of prescribed burning on infiltration and sediment production in the pinyon-juniper woodland, Nevada. J. Range Manage., 31(4), 250-253.

Simanton, J. R., Weltz, M. A., & Larsen, H. D. (1991), Rangeland experiments to parameterize the water erosion prediction project model: vegetation canopy cover effects.

Journal of Range Management, 44(3), 276-282.

Stone, J. J., Paige, G. B., & Hawkins, R. H. (2008), Rainfall intensity-dependent infiltration rates on Rangeland rainfall simulator plots. Transactions of the ASABE, 51(1), 45-53.

6. By inspecting the data sets available for downloading, I saw that many of the experiments were restricted to 45 minutes (e.g. small_time_series-csv). May the authors explain such time restriction?

Authors' Response: The duration for each rainfall simulation was set to 45 min for experimental and logistical purposes. The authors have extensive experience conducting rainfall simulations (for example see Pierson et al., 2001, 2002a, 2002b, 2008a, 2008b, 2009). Based on that experience, we found that steady state infiltration and runoff generally occur within 45 min for most rainfall simulation applications at moderate to high rainfall intensities. Of course, infiltration for a given rainfall intensity varies with soil properties, surface conditions, and vegetation cover. We anticipated the 45 min duration would allow enough time for steady state infiltration and runoff on most of our plots, particularly for the highest intensity. Steady state infiltration and runoff were not always achieved with our design, and, in some cases, no runoff occurred. This is common for experiments that span the variability in conditions encountered in our experiments. The 45 min duration was also selected so that we could achieve the required replications across the various Site $\times$ Treatment $\times$ Microsite combinations each field season. The selected duration is similar to durations used in numerous other rainfall simulation studies (see short list of studies cited below and in comment #5 above for example), typically in the range of 30 min to 60 min. There is no need to state such a justification in the manuscript given our approach is typical for rainfall simulation experiments and that the duration is provided in the methods description.

Pierson, F. B., Bates, J. D., Svejcar, T. J., & Hardegree, S. P. (2007), Runoff and erosion after cutting western juniper. Rangeland Ecology and Management, 60(3), 285-292.

Pierson, F. B., Carlson, D. H., & Spaeth, K. E. (2002a), Impacts of wildfire on soil hydrological properties of steep sagebrush-steppe rangeland. International Journal of Wildland Fire, 11(2), 145-151.

Pierson, F. B., Moffet, C. A., Williams, C. J., Hardegree, S. P., & Clark, P. E. (2009), Prescribed-fire effects on rill and interrill runoff and erosion in a mountainous sage-brush landscape. Earth Surface Processes and Landforms, 34(2), 193-203. doi: 10.1002/esp.1703.

Pierson, F. B., Robichaud, P. R., Moffet, C. A., Spaeth, K. E., Hardegree, S. P., Clark, P. E., & Williams, C. J. (2008a), Fire effects on rangeland hydrology and erosion in a steep sagebrush-dominated landscape. Hydrological Processes, 22(16), 2916-2929. doi: 10.1002/hyp.6904.

Pierson, F. B., Robichaud, P. R., Moffet, C. A., Spaeth, K. E., Williams, C. J., Hardegree, S. P., & Clark, P. E. (2008b), Soil water repellency and infiltration in coarse-textured soils of burned and unburned sagebrush ecosystems. Catena, 74, 98-108.

Pierson, F. B., Robichaud, P. R., & Spaeth, K. E. (2001), Spatial and temporal effects of wildfire on the hydrology of a steep rangeland watershed. Hydrological Processes, 15(15), 2905-2916. doi: 10.1002/hyp.381.

Pierson, F. B., Spaeth, K. E., Weltz, M. A., & Carlson, D. H. (2002b), Hydrologic response of diverse western rangelands. Journal of Range Management, 55(6), 558-570.

7. Lastly, while I highly appreciate the efforts the authors put into the generation of this data set, I was wondering how these data relate to previous studies conducted in other study areas but the ones presented here. Do the authors see the chance to use and/or transfer their data set for studies outside the Great Basin area?

Authors' Response: The data presented in this manuscript were collected in support of research on conifer encroachment and various practices to arrest tree advance

and infill in Great Basin sagebrush steppe. Woody plant encroachment is occurring on water-limited sparsely-vegetated landscapes around the World. Typically, as woody plants encroach, herbaceous vegetation declines, the plant community structure coarsens, and connectivity of bare ground and runoff and sediment sources increases (Schlesinger et al., 1990; Wainwright et al., 2000; Turnbull et al., 2009, 2012; Williams et al., 2014). These changes commonly result in elevated runoff and erosion rates and long-term loss of ecologically important surface soil. Without management intervention or natural disturbance, such plant community transitions can become self-perpetuating (Turnbull et al., 2012). These structural and functional relationships are consistent with woodland encroachment effects on the sites in this dataset and with the ecohydrologic responses to management that the dataset spans (e.g., Pierson et al., 2010; Williams et al., 2014, 2016, 2019a, 2019b, 2020). Trends in runoff and erosion rates associated with wildfire and land use induced changes in vegetation, groundcover, and soils generally follow similar trends as those across our dataset (Cerdà and Doerr, 2005; Ludwig et al., 2005, 2007; Turnbull et al., 2010; Moody et al., 2013). Given these fundamental relationships, we anticipate that our dataset is applicable in forecasting potential relative changes in runoff and erosion under similar plant community dynamics. It is intuitive that actual runoff and erosion rates and vegetation responses to treatments may vary for different climate, soil, topographic, and other site-specific attributes in other domains. The dataset also transfers for use in evaluating/validating predictive capability of and potentially enhancing runoff and erosion models developed for water-limited lands such as rangeland and woodlands on sloping topography (e.g., Al-Hamdan et al., 2012, 2015). The points presented here are retained from the original Abstract (now at Lines 31-39) and Summary and Conclusions (now at Lines 470-481).

Al-Hamdan, O. Z., Hernandez, M., Pierson, F. B., Nearing, M. A., Williams, C. J., Stone, J. J., Boll, J., & Weltz, M. A. (2015), Rangeland Hydrology and Erosion Model (RHEM) enhancements for applications on disturbed rangelands. Hydrological Processes, 29(3), 445-457. doi: 10.1002/hyp.10167.

Al-Hamdan, O. Z., Pierson, F. B., Nearing, M. A., Williams, C. J., Stone, J. J., Kormos, P. R., Boll, J., & Weltz, M. A. (2012), Concentrated flow erodibility for physically based erosion models: Temporal variability in disturbed and undisturbed rangelands. Water Resources Research, 48(7). doi: 10.1029/2011WR011464.

Cerdà, A., & Doerr, S. H. (2005), Influence of vegetation recovery on soil hydrology and erodibility following fire: An 11-year investigation. International Journal of Wildland Fire, 14(4), 423-437.

Ludwig, J. A., Bartley, R., Hawdon, A. A., Abbott, B. N., & McJannet, D. (2007), Patch configuration non-linearly affects sediment loss across scales in a grazed catchment in north-east Australia. Ecosystems, 10(5), 839-845. doi: 10.1007/s10021-007-9061-8.

Ludwig, J. A., Wilcox, B. P., Breshears, D. D., Tongway, D. J., & Imeson, A. C. (2005), Vegetation patches and runoff-erosion as interacting ecohydrological processes in semiarid landscapes. Ecology, 86(2), 288-297.

Moody, J. A., Shakesby, R. A., Robichaud, P. R., Cannon, S. H., & Martin, D. A. (2013), Current research issues related to post-wildfire runoff and erosion processes. Earth-Science Reviews, 122, 10-37. doi: 10.1016/j.earscirev.2013.03.004.

Pierson, F. B., Williams, C. J., Kormos, P. R., Hardegree, S. P., Clark, P. E., & Rau, B. M. (2010), Hydrologic vulnerability of sagebrush steppe following pinyon and juniper encroachment. Rangeland Ecology and Management, 63(6), 614-629. doi: 10.2111/rem-d-09-00148.1.

Schlesinger, W. H., Reynolds, J. F., Cunningham, G. L., Huenneke, L. F., Jarrell, W. M., Virginia, R. A., & Whitford, W. G. (1990), Biological feedbacks in global desertification. Science, 247(4946), 1043-1048.

Turnbull, L., Wainwright, J., & Brazier, R. E. (2008), A conceptual framework for understanding semi-arid land degradation: ecohydrological interactions across multiple-space and time scales. Ecohydrology, 1(1), 23-34.

Turnbull, L., Wainwright, J., Brazier, R. E., & Bol, R. (2010), Biotic and Abiotic Changes in Ecosystem Structure over a Shrub-Encroachment Gradient in the Southwestern USA. Ecosystems, 13(8), 1239-1255. doi: 10.1007/s10021-010-9384-8.

Turnbull, L., Wilcox, B. P., Belnap, J., Ravi, S., D'Odorico, P., Childers, D., Gwenzi, W., Okin, G., Wainwright, J., Caylor, K. K., & Sankey, T. (2012), Understanding the role of ecohydrological feedbacks in ecosystem state change in drylands. Ecohydrology, 5(2), 174-183. doi: 10.1002/eco.265.

Wainwright, J., Parsons, A. J., & Abrahams, A. D. (2000), Plot-scale studies of vegetation, overland flow and erosion interactions: case studies from Arizona and New Mexico. Hydrological Processes, 14(16-17), 2921-2943.

Williams, C. J., Pierson, F. B., Al-Hamdan, O. Z., Kormos, P. R., Hardegree, S. P., & Clark, P. E. (2014), Can wildfire serve as an ecohydrologic threshold-reversal mechanism on juniper-encroached shrublands. Ecohydrology, 7(2), 453-477. doi: 10.1002/eco.1364.

Williams, C. J., Pierson, F. B., Kormos, P. R., Al-Hamdan, O. Z., Nouwakpo, S. K., & Weltz, M. A. (2019a), Vegetation, Hydrologic, and erosion responses of sagebrush steppe 9 yr following mechanical tree removal. Rangeland Ecology and Management, 72(1), 47-68. doi: 10.1016/j.rama.2018.07.004.

Williams, C. J., Pierson, F. B., Nouwakpo, S. K., Al-Hamdan, O. Z., Kormos, P. R., & Weltz, M. A. (2020), Effectiveness of prescribed fire to re-establish sagebrush steppe vegetation and ecohydrologic function on woodland-encroached sagebrush rangelands, Great Basin, USA: Part I: vegetation, hydrology, and erosion responses. Catena, 185. doi: 10.1016/j.catena.2018.02.027.

Williams, C. J., Pierson, F. B., Nouwakpo, S. K., Kormos, P. R., Al-Hamdan, O. Z., & Weltz, M. A. (2019b), Long-term evidence for fire as an ecohydrologic threshold-reversal mechanism on woodland-encroached sagebrush shrublands. Ecohydrology,
12(4). doi: 10.1002/eco.2086.

Williams, C. J., Pierson, F. B., Robichaud, P. R., Al-Hamdan, O. Z., Boll, J., & Strand, E. K. (2016), Structural and functional connectivity as a driver of hillslope erosion following disturbance. International Journal of Wildland Fire, 25(3), 306-321.

RESPONSES TO ANONYMOUS REFEREE #2 COMMENTS:

8. The manuscript presents extensive data on numerous parameters characterizing surface and shallow subsurface hydrology at three locations within the western U.S. These data are concise and relevant for future hydrological and sedimentary analysis, and potential inclusion to various land surface models. The manuscript is available for download via the URL provided by the authors.

Authors' Response: We appreciate Reviewer 2's comments here regarding the extensiveness and relevancy of the dataset.

9. The description of plot scales should be consistent throughout the manuscript. In the Abstract, only 'overland flow' plots are mentioned explicitly; this changes to rainfall simulations at various plot sizes and overland flow plots in Lines 111-113, and finally to four plot scales in Lines 148-150, hillslope plots added. Besides, a small figure showing locations for each plot could be useful for non-U.S. readership. This inconsistency is brought further to the text, Section 3, where field methods description starts with hillslope-scale plots, the largest, and continues with small- and large-scale plots etc. Though there might be a certain logic in such description order, I would suggest to follow either top-down or bottom-up approach.

Authors' Response: We appreciate Reviewer 2 bringing this to our attention. We believe the bulk of the confusion is associated with the aforementioned omissions of labels (showing various plot scales) in Table 2 (see response to Reviewer 1 comment #2 above). We have corrected Table 2 to show the labels, as discussed in above comment #2 response, and that revision should provide clarity regarding the various plot scales.

As for the abstract, we initially avoided specific information on various plot scales to simply focus on processes, which have a scale dependency. We considered methods in the abstract abbreviated, but added detail given the issue presented. To address confusion, we added specific details to the abstract text regarding the various plots with runoff and erosion measurements, following the top-down approach (at Lines 27-31):

"The methodologies applied in data collection and the cross-scale experimental design uniquely provide scale-dependent, separate measures of interrill (rainsplash and sheetflow processes, 0.5 m2 plots) and concentrated overland-flow runoff and erosion rates (∼9 m2 plots), along with collective rates for these same processes combined over the patch scale (13 m2 plots)."

At lines 148-150, we clearly specify each of the plot types and the respective scales in a top-down model as suggested by the reviewer, and follow that text with basic experimental design presentation and explanation of what is measured at each plot scale throughout the rest of the section (with multiple references to corrected Table 2). We did re-arrange the paragraph text to ensure the text follows the scale presentation of the opening sentence, Lines 148-150 that read:

"A suite of biological and physical attributes at each site were measured at point, small-rainfall plot (0.5 m2), overland-flow plot (∼9 m2), large-rainfall plot (13 m2), and hillslope plot (990 m2) scales."

The "Field Methods" section (Section 3) provides the explanation of sampling methods by plot type. There, we do begin with the hillslope scale plots simply because those only include vegetation and ground cover measures. All of the other plot types include vegetation, ground cover, soil, and hydrology/erosion measures. Also, there is some practical groupings of the small plot and large plot rainfall simulations (due to similarities in methods across the scales) and then presentation of the overland flow methods. This methodological presentation was used in nearly all of the published papers (15+)

on the dataset and was retained for continuity with those papers. This may be particularly useful if a user is going back to these papers for more specific details on the various methods. This presentation also clearly separates the various methodologies by respective plot scales.

The corrections to Table 2 (see response to comment #2 above for Reviewer 1), amendments to the abstract noted above, and clear description of the various plot scales in Section 2 (Lines 148-150) remedy the issue presented here by Reviewer 2. Additionally, we have made multiple minor text insertions to clarify measurement scales in various areas of the manuscript.

Reviewer 2 also suggests a figure showing the various plot locations, but the number of plots across the sites, study years, treatments, etc. would be very cumbersome for a reader (too many symbols, etc.). However, we do agree that adding the site locations would potentially be helpful for a non-US reader. We have added latitude and longitude information in for each site, underneath the site names/locations in Table 1. A reader can easily enter these numbers into Google Earth or another mapping software to see the study site locations and visualize the sites to the degree possible by the selected software.

10. Lines 287-288, the sediment concentration is said to be calculated from runoff samples by weighing; what is a 'runoff sample'? Is it a liquid volume - and if yes, was it just dried to full sample evaporation? If not, was any filtration system used, and if yes, then what were its parameters - pore size etc?

Authors' Response: The runoff samples were indeed liquid samples as is generally intuitive for runoff samples. Each sample was collected in the field in a numbered sample bottle and was retained in that sealed bottle for processing at the laboratory. Each runoff sample (water, sediment, and bottle) was weighed in the laboratory and the mass was recorded. Each bottle (with all water and sediment retained) was then placed in an oven set at 105° C and left in the oven until all water was evaporated. Each

bottle was then removed from the oven, reweighed, and the remaining mass (sample bottle and sediment) was recorded. Each bottle was then washed of all sediment, air dried, and then weighed to determine the bottle tare mass. For each sample, the mass of water from the original runoff sample was calculated by subtracting the respective mass of the dry sediment and sample bottle from the combined total mass of the water, sediment, and sample bottle. Likewise, the sediment mass for each sample was calculated by subtracting the respective sample bottle tare mass from the measured mass of the respective dry sediment and sample bottle. Runoff samples were not filtered at any stage of laboratory processing. The above described methodology is considered a standard laboratory procedure for these types of experiments and is more simply described by the current text in the manuscript, with exception perhaps of the lack of filtering. Filtering is sometimes used to reduce sample drying times, but we did not employ this method. Our current statement regarding processing of samples is typical for runoff sample processing and is a generally accepted statement for publication given the standard methods. However, we have now clarified samples were not filtered. The full text referenced here by Reviewer 2 and the addition of new text on filtering now reads (at Lines 287-289):

"Cumulative runoff and sediment amounts were obtained for each runoff sample by weighing the sample before and after drying at 105°C (Pierson et al., 2010). Runoff samples were not filtered at any stage of laboratory processing."

11. The dataset is well-organized, but several technical corrections are needed:

Authors' Response: Each of the items presented by Reviewer 2 are addressed in the responses below (comments #12-#18).

12. Section 3.1. Data Dictionary - data types should be presented as standard notation, i.e. integer, real, character etc; same, variable sizes should be given, i.e. as INT/LONG INT/DOUBLE/CHAR(X) etc.

Authors' Response: We appreciate Reviewer 2's comment here regarding the data

structure and considered recoding the data structure, including the associated variable items. In short, we retained the original data structure in the form required by the approved data repository to minimize confusion across an array of potential end users, as explained here. The final dataset was organized by the authors and submitted to and reviewed by the US Department of Agriculture, National Agricultural Library, Ag Data Commons (https://data.nal.usda.gov/), an approved data repository for ESSD datasets. The submission was subjected to the requirements by the data repository and the final dataset meets all requirements of the data repository, including the data dictionary and data structure. The dataset has been archived by the data repository and has been assigned a doi (https://doi.org/10.15482/USDA.ADC/1504518). As such, the dataset and its structure have been approved, established, and archived by the data repository in a commonly accepted format. The suggestions here by Reviewer 2 are indeed also common, but are somewhat a component of data application associated with one of an array of potential end users and applications. The needs of end users vary extensively depending on the data application and software applied, as such meeting all potential desired structures is impractical. The dataset items noted here by Reviewer 2 and others noted by Reviewer 2 in subsequent comments below (see comments #13, #14, #17, and #18 below) are all easily addressed by an end user through some simple recoding, typical in downloading and using any dataset. We have elected to retain our approved and archived data structure, per the data repository, in lieu of developing amended versions that may further add confusion associated with archiving duplicative tables and data structures of the same dataset. We see the potential confusion induced by duplicative tables as being more confusing for end users than having the one existing data structure relative to many other possible structures. Further, the data structure is consistent with another recent similar dataset published by ESSD, Polyakov et al. (2018).

Polyakov, V., Stone, J., Holifield Collins, C., Nearing, M. A., Paige, G., Buono, J., and Gomez-Pond, R.-L. (2018), Rainfall simulation experiments in the southwestern USA using the Walnut Gulch Rainfall Simulator. Earth Syst. Sci. Data, 10, 19–26. doi:

10.5194/essd-10-19-2018.

13. Section 3.2. Categorical variables are multiple in the Data Dictionary, and are particularly poorly described; possible categories are listed as 'Acceptable values', which is not the best way to present them. No explanation on whet does, e.g. 'Tracked_LowMulch' mean, is given in the dataset itself. A separate table explaining your categorical variables is needed, or you might suggest a better way of presentation.

Authors' Response: Please see our response to Reviewer 2 comment #12 above, which also applies in full to this comment.

14. Section 3.3. Same, 'Yes/No' is not a character variable, but has LOGICAL type, therefore acceptable values are 0/1, Y/N, or T/F, each is valid.

Authors' Response: Please see our response to Reviewer 2 comment #12 above, which also applies in full to this comment.

15. Section 3.4. Dataset contains some info on treatment area and date, but I've found no clear descriptors for treatment type for each dataset in the plot characteristics table. This raises the question on whether the variables are correctly distributed between various dataset tables.

Authors' Response: The authors do not understand the meaning of this comment. Each of the sub-datasets contain the information for treatment/treatment area, treated (yes or no), and treatment date, as explained here. Table 2 includes a column for treatment for each plot type and shows the number of plots sampled in each treatment for each Site and Site × Microsite combination. All other data tables presented in the paper (Tables 3 and 5-11), with exception of the soil texture and bulk density data table (Table 4), include columns for Treatment/Treatment Area, Treated Yes or No, and Year. The respective tables in the data repository, https://doi.org/10.15482/USDA.ADC/1504518, all contain columns for Treatment/Treatment Area, Treated Yes or No, and Treatment Date. All data and the data

structure were evaluated extensively by the authors and the data repository prior to submission and approval for posting by the data repository, as explained in the response to Reviewer 2 comment #12 above.

16. Section 3.5. Table 3 contains no info on either plot type (small vs large vs overland etc) or plot area.

Authors' Response: The confusion here stems in part from the lack of labels on Table 2 for the various plot types. Table 3 shows data for the hillslope-scale site characterization plots (990 m2). The table caption does show the plot type and plot area, contrary to the reviewer comment here, and explains that the data are foliar and ground cover measures. The revised Table 2, showing the associated labels provides additional clarity in addressing this issue (see response to Reviewer 1, comment #2 above).

17. Section 3.6. I find it difficult to browse through data with visual inspection, since: PLOT_ID is a last column, e.g. in Table 4, and is hard to find in other tables as well; in several tables, PLOT_ID is not unique since two rows contain data for differerent years; treatment date repeats in Tables 3 and 4.

Authors' Response: Please see our response to Reviewer 2 comment #12 above, which also applies in full to this comment.

18. In general, column sequence is not entirely logical, and can be enhanced. The dataset structure, I believe, should be subject to technical inspection. I suggest the authors to read your dataset to R/RStudio environment and check dataset usability / statistical analysis performance.

Authors' Response: Please see our response to Reviewer 2 comment #12 above, which also applies in full to this comment. The data have undergone technical inspection as part of the data repository requirements. There are many different data structures possible to suit various potential data applications and software tools. Multiple data structures (i.e., multiple versions of tables) to accommodate all possible software

applications is not merited and perhaps induces more confusion for end users. Some data reorganization to meet end user needs is typical with data extraction from repositories and is readily accomplished through simple coding in various data management software packages, including R.

RESPONSES TO ANONYMOUS REFEREE #3 COMMENTS:

19. General comments: Authors present extensive and detailed dataset with vegetation, ground cover, soils, hydrology, and erosion data from over 1000 plots in diverse vegetation, ground cover, and surface soil conditions from three study sites in USA for five study years. Presented data is of high scientific importance and probable usage in the future. Study sites, experimental design and field methods are well described.

Authors' Response: The authors thank Reviewer 3 for these comments regarding the extensiveness, scientific importance, and utility of the dataset.

20. There are no explicit estimates of the data errors and its discussion. Consider adding some uncertainty estimates in the Section 2 or Section 3.

Authors' Response: The best estimates of error for the type of measures presented would simply be indicated by variability for each measure. We have elected rather to provide the actual data and allow users to make such evaluations regarding application. The data have been well published in various papers (15+ published papers) as cited in the reference section for the manuscript. As such, assessments that include those measures of variability are readily available through other publications. Our intent here is not to re-analyze these data, but rather to provide the data in full form for use by others. Of course, that assumes end users will make their own assessment on the utility of the dataset for the desired application.

21. Paper does not provide information about which exactly kind of data is in the dataset. Reader is not able to decide whether he/she interested to download data or not based just on the paper. I suggest including a new section or subsection or

extend Section 5 and include brief technical overview of the data covering description of variables from the dataset (maybe in a table that is shorter version of the table "SageSTEP_Database_Data_Dictionary" from the dataset), technical details (could be from lines 450-461) and structure of the data files.

Authors' Response: Although we appreciate the comment here, we opine this is unnecessary. The dataset has been well published and context for the dataset is well explained in the abstract. There is also a more detailed description of the dataset at the required data repository, https://doi.org/10.15482/USDA.ADC/1504518. We originally included much of that more detailed description in this paper, but were required by the journal editors to reduce that duplicative content. Given the numerous publications from the dataset (15+ papers, see references), the description already available at the data repository, and the abstract here, we do not see clear merit of adding an additional summary as suggested here by the reviewer.

22. Section 4 is important for understanding of scientific significance of the presented dataset but lacks any scientific conclusions. It explains the previous usage of data. It would be good for readers to know not only descriptions of data usage but also the scientific results. I suggest expanding the section, brief presenting significant findings of the mentioned studies and referring to the Figures 3-5.

Authors' Response: See response to comment #21 above in this regard. Results from the various data collection studies for the greater dataset presented are well published already in a series of 15+ papers. Repeating those here is duplicative and unnecessary. The series of papers published to date on this dataset span pre-treatment conditions (Pierson et al., 2010, 2013; Williams et al., 2014), initial impacts of tree removal treatments (Cline et al., 2010; Pierson et al., 2014, 2015; Williams et al., 2014, 2016), longer-term impacts of tree removal (Nouwakop et al., 2020; Williams et al., 2019a, 2020), and a full analysis spanning pre-treatment, short-term responses, and long-term responses for tree removal by fire (Williams et al., 2019b). The dataset application in development of hydrology and erosion model parameters has been well

published in a suite of papers by Al-Hamdan et al. (2012a, 2012b, 2013, 2015, 2017). Additionally, a manuscript of the research findings spanning the entire study and with additional measurements on long-term soil erosion rates and 13 yr treatment effects is in preparation for submission later this year. Our goal for the current manuscript is simply to provide a basic description of the study, the methods, and the available dataset (with linkage to the required repository), along with some abbreviated presentation on data uses (current Section 4). It is not our intent to re-present analyses and results here, as they stand alone already in the various publications, see list below.

Al-Hamdan, O. Z., Hernandez, M., Pierson, F. B., Nearing, M. A., Williams, C. J., Stone, J. J., Boll, J., & Weltz, M. A. (2015), Rangeland Hydrology and Erosion Model (RHEM) enhancements for applications on disturbed rangelands. Hydrological Processes, 29(3), 445-457. doi: 10.1002/hyp.10167.

Al-Hamdan, O. Z., Pierson, F. B., Nearing, M. A., Stone, J. J., Williams, C. J., Moffet, C. A., Kormos, P. R., Boll, J., & Weltz, M. A. (2012a), Characteristics of concentrated flow hydraulics for rangeland ecosystems: implications for hydrologic modeling. Earth Surface Processes and Landforms, 37(2), 157-168. doi: 10.1002/esp.2227.

Al-Hamdan, O. Z., Pierson, F. B., Nearing, M. A., Williams, C. J., Hernandez, H., Boll, J., Nouwakpo, S. K., Weltz, M. A., & Spaeth, K. E. (2017), Developing a parameterization approach for soil erodibility for the Rangeland Hydrology and Erosion Model (RHEM). Transactions of the American Society of Agricultural and Biological Engineers, 60(1), 85-94. doi: 10.13031/trans.11559.

Al-Hamdan, O. Z., Pierson, F. B., Nearing, M. A., Williams, C. J., Stone, J. J., Kormos, P. R., Boll, J., & Weltz, M. A. (2012b), Concentrated flow erodibility for physically based erosion models: temporal variability in disturbed and undisturbed rangelands. Water Resources Research, 48(7), W07504.

Al-Hamdan, O. Z., Pierson, F. B., Nearing, M. A., Williams, C. J., Stone, J. J., Kormos, P. R., Boll, J., & Weltz, M. A. (2013), Risk assessment of erosion from concentrated flow

on rangelands using overland flow distribution and shear stress partitioning. Transactions of the ASABE, 56(2), 539-548.

Cline, N. L., Roundy, B. A., Pierson, F. B., Kormos, P., & Williams, C. J. (2010), Hydrologic response to mechanical shredding in a juniper woodland. Rangeland Ecology and Management, 63(4), 467-477.

Nouwakpo, S. K., Williams, C. J., Pierson, F. B., Weltz, M. A., Arslan, A., & Al-Hamdan, O. Z. (2020), Effectiveness of prescribed fire to re-establish sagebrush steppe vegetation and ecohydrologic function on woodlandencroached sagebrush rangelands, Great Basin, USA: Part II: Runoff and sediment transport at the patch scale. Catena, 185, 104301. doi: 10.1016/j.catena.2019.104301.

Pierson, F. B., Williams, C. J., Hardegree, S. P., Clark, P. E., Kormos, P. R., & Al-Hamdan, O. Z. (2013), Hydrologic and erosion responses of sagebrush steppe following juniper encroachment, wildfire, and tree cutting. Rangeland Ecology and Management, 66(3), 274-289.

Pierson, F. B., Williams, C. J., Kormos, P. R., & Al-Hamdan, O. Z. (2014), Short-term effects of tree removal on infiltration, runoff, and erosion in woodland-encroached sagebrush steppe. Rangeland Ecology and Management, 67(5), 522-538. doi: 10.2111/rem-d-13-00033.1.

Pierson, F. B., Williams, C. J., Kormos, P. R., Al-Hamdan, O. Z., Hardegree, S. P., & Clark, P. E. (2015), Short-term impacts of tree removal on runoff and erosion from pinyon- and juniper-dominated sagebrush hillslopes. Rangeland Ecology and Management, 68(5), 408-422. doi: 10.1016/j.rama.2015.07.004.

Pierson, F. B., Williams, C. J., Kormos, P. R., Hardegree, S. P., Clark, P. E., & Rau, B. M. (2010), Hydrologic vulnerability of sagebrush steppe following pinyon and juniper encroachment. Rangeland Ecology and Management, 63(6), 614-629. doi: 10.2111/rem-d-09-00148.1.

Williams, C. J., Pierson, F. B., Al-Hamdan, O. Z., Kormos, P. R., Hardegree, S. P., & Clark, P. E. (2014), Can wildfire serve as an ecohydrologic threshold-reversal mechanism on juniper-encroached shrublands? Ecohydrology, 7(2), 453-477. doi: 10.1002/eco.1364.

Williams, C. J., Pierson, F. B., Kormos, P. R., Al-Hamdan, O. Z., Nouwakpo, S. K., & Weltz, M. A. (2019a), Vegetation, hydrologic, and erosion responses of sagebrush steppe 9 yr following mechanical tree removal. Rangeland Ecology and Management, 72(1), 47-68. doi: 10.1016/j.rama.2018.07.004.

Williams, C. J., Pierson, F. B., Nouwakpo, S. K., Al-Hamdan, O. Z., Kormos, P. R., & Weltz, M. A. (2020), Effectiveness of prescribed fire to re-establish sagebrush steppe vegetation and ecohydrologic function on woodland-encroached sagebrush range-lands, Great Basin, USA: Part I: vegetation, hydrology, and erosion responses. Catena, 185, 103477. doi: 10.1016/j.catena.2018.02.027.

Williams, C. J., Pierson, F. B., Nouwakpo, S. K., Kormos, P. R., Al-Hamdan, O. Z., & Weltz, M. A. (2019b), Long-term evidence for fire as an ecohydrologic threshold-reversal mechanism on woodland-encroached sagebrush shrublands. Ecohydrology, 12(4). doi: 10.1002/eco.2086.

Williams, C. J., Pierson, F. B., Robichaud, P. R., Al-Hamdan, O. Z., Boll, J., & Strand, E. K. (2016), Structural and functional connectivity as a driver of hillslope erosion following disturbance. International Journal of Wildland Fire, 25(3), 306-321. doi: 10.1071/WF14114.

23. Table 1: Intercanopy bare ground includes shrubs and grasses?

Authors' Response: We appreciate the reviewer pointing this out. "Intercanopy" refers to the area between tree canopies consisting of shrubs, grasses, and interspaces between plants (i.e., shrub-interspace zone). So, Reviewer 3 is correct here, that intercanopy bare ground should not include shrubs and grasses. To correct the error, we

have added the text "Intercanopy refers to the…." at the beginning of the footnote referenced by "Intercanopy bare ground (%)11", which now reads:

"Intercanopy refers to the area between tree canopies consisting of shrubs, grasses, and interspaces between plants (i.e., shrub-interspace zone)."

24. Table 2: There are 4 parts of the Table. What do they refer to? Consider adding informative titles to different parts of the table and relocate extensive description of different types of sites to the paper text.

Authors' Response: Please see response to Reviewer 1 comment #2 above, which addresses this issue. We thank Reviewer 3 for pointing this out. We have made the corrections to Table 2 as indicated in response to comment #2.

25. Line 206: Are site characterization plots representative for all plots at each of three study sites?

Authors' Response: The authors are not exactly sure of the question here, as the measures for these plots are presented in Section 3.1. The site characterization plots provide measures of hillslope scale vegetation and ground cover in each treatment area at the sites prior to treatments (2006) and in each treatment area at the sites 1 yr post-treatment (2007) and 9 yr post-treatment (2015). Only site characterization data for Marking Corral and Onaqui are shown, see Table 3. The corrections to Table 2 may also alleviate this question (see response to Reviewer 1 comment #2 above).

26. Lines 450-459: Consider to relocate this detailed description of the dataset from the Conclusions to Section 5 or new Section / subsection with the technical overview of the data.

Authors' Response: Please see response to Reviewer 3 comment #22 above.

27. Data table "Small time series": Please explain what empty cells mean, for example lines No 6099, 7431, 7504, 8349 of the columns "Runoff_L_min", "SedConc_g_L", "Runoff_mm_hr" and "SedDisch_g_s".

Authors' Response: These are cases in which the runoff sample was discarded due to laboratory or field errors (e.g., bottle spillage). We will work with the data repository to determine the best way to re-code (as missing) or remove these lines for this time series data file, if necessary.

28. Link to the data DOI in the abstract and Section 5 leads to DOI Not Found webpage.

Authors' Response: The authors confirmed the link is active and correct. Perhaps there was a temporary outage at the data repository or in the user network at the time access was tested by Reviewer 3.

29. Line 387-389: It would be useful to show TRAW and width variables on the photo or on the scheme.

Authors' Response: We can understand the utility of such a photo, but find it difficult to clearly identify the full wetted width and individual flow paths widths in the photos as measured at cross-sections 1 m, 2 m, and 3 m downslope from the flow release. However, we provided a detailed diagram of these measures in an earlier paper (Pierson et al., 2008) and have added reference to that paper. The diagram there should provide clarity on the methods without replication of the figure in this publication (which we are trying to avoid per the editorial staff). The text at the noted location now reads, at Lines 388-391:

"The width, depth, and a total rill area width (TRAW) of overland flow were measured along flow cross-sections 1 m, 2 m, and 3 m downslope from the flow release point (Pierson et al., 2010). The TRAW variable represents the total width between the outermost edges of the outermost flow paths at the respective cross section (see Pierson et al., 2008)."

30. Figure 3: Do (a) and (c) refer to Marking Corral site and (b) and (d) – to Onaqui site? It should be explicitly noted in the Figure caption.

Authors' Response: Reviewer 3 is correct here regarding the figure assignments. We

have amended the figure caption, as shown below, to clarify the figure assignments:

"Figure 3. Example infiltration (a [Marking Corral] and b [Onaqui]), calculated as applied rainfall minus measured runoff, and sediment discharge (c [Marking Corral] and d [Onaqui]) time series data generated from a subset of the small-plot rainfall simulation dataset. Example sub-dataset is from wet-run rainfall simulations in untreated (Cont) and burned (Burn) interspace (Int), shrub coppice (Shr), and tree coppice (Tree) microsites at the Marking Corral and Onaqui study sites 9 yr following prescribed fire. The data illustrate the long-term impacts of burning and associated changes in surface conditions on infiltration and sediment discharge. Figure modified from Williams et al. (2020)."

31. Untreated tree coppice microsite indicated as bold green line in the legend but dash line on the graph. It would be better to use bold lines for all three control microsites.

Authors' Response: Reviewer 3 is referring to the lines in Figure 3 and is correct in regards to the error. We have corrected the figure legend to correctly identify each of the lines drawn in the figure. We elected to retain the dash format. Users that print in black and white may need the line variations to correctly separate one line from another. Using all solid lines for controls would hinder such separation in a black and white version.

32. Table 5-6: expand abbreviations Fol. Cvr., JUOC and WDPT.

Authors' Response: We reviewed all tables for abbreviation issues and addressed those that were not intuitive. These abbreviations are explained in the data dictionary at the data repository, but we provide them now in the respective table captions for these abbreviated tables. The captions have been revised as shown below:

Table 4. Soil texture and bulk density variables and data structure for those measures for all study sites. Abbreviations in the table example are as follows: juniper_cop refers to juniper coppice microsites; shrub_cop refers to shrub coppice microsites; and

pinyon_cop refers to pinyon coppice microsites.

Table 5. Example (subset) of vegetation and ground cover variables and data structure for measures on hillslope-scale site characterization plots (990 m2) at the study sites. Abbreviations in the table example are as follows: Fol. Cvr. refers to Foliar Cover; and JUOC refers to western juniper (Juniperus Occidentalis Hook.).

Table 6. Example (subset) of rainfall simulation, vegetation, ground cover, and soil variables and data structure for measures on small-rainfall simulation plots (0.5 m2) at the study sites. Abbreviations in the table example are as follows: Fol. Cvr. refers to Foliar Cover; Grd. Cvr. refers to Ground Cover; WDPT refers to Water Drop Penetration Time; shrub_cop refers to shrub coppice microsites; pinyon_cop refers to pinyon coppice microsites; and juniper_cop refers to juniper coppice microsites.

Table 7. Example (subset) of rainfall simulation, vegetation, ground cover, and soil variables and data structure for measures on large-rainfall simulation plots (13 m2) at the study sites. Abbreviations in the table example are as follows: Fol. Cvr. refers to Foliar Cover; Grd. Cvr. refers to Ground Cover; Avg. refers to average; juniper_cop refers to juniper coppice microsites; and pinyon_cop refers to pinyon coppice microsites.

Table 8. Example (subset) of overland flow, vegetation, and ground cover variables and data structure for measures on overland flow simulation plots (∼9 m2) at the study sites. Abbreviations in the table example are as follows: Avg. refers to average; juniper_cop refers to juniper coppice microsites; and pinyon_cop refers to pinyon coppice microsites.

Table 9. Example (subset) of time series runoff and sediment data from small-plot rainfall simulations (0.5 m2) at the study sites. Abbreviations in the table example are as follows: Conc. refers to concentration; and shrub_cop refers to shrub coppice microsites.

Table 10. Example (subset) of time series runoff and sediment data from large-plot

rainfall simulations (13 m2) at the study sites. Abbreviations in the table example are as follows: Conc. refers to concentration; and juniper_cop refers to juniper coppice microsites.

Table 11. Example (subset) of time series runoff and sediment data from overland flow simulations (∼9 m2) at the study sites. Abbreviations in the table example are as follows: Conc. refers to concentration; and juniper_cop refers to juniper coppice microsites.